# Impact of obstetric unit closures, travel time and distance to obstetric services on maternal and neonatal outcomes in high-income countries: a systematic review

Reem Saleem Malouf ®,[1] Claire Tomlinson,[2] Jane Henderson,[1] Charles Opondo ®,[1] Peter Brocklehurst,[1] Fiona Alderdice,[1] Angaja Phalguni,[2] Janine Dretzke[2]

[1]Nuffield Department of Population Health, Policy Research Unit in Maternal Health and Care, National Perinatal Epidemiology Unit, University of Oxford, Oxford, UK
[2]Institute of Applied Health Research, College of Medical and Dental Sciences, University of Birmingham, Birmingham, UK

**Correspondence to**
Dr Reem Saleem Malouf;
reem.malouf@npeu.ox.ac.uk

## ABSTRACT

**Objectives** To systematically review (1) The effect of obstetric unit (OU) closures on maternal and neonatal outcomes and (2) The association between travel distance/time to an OU and maternal and neonatal outcomes.

**Design** Systematic review of any quantitative studies with a comparison group.

**Data sources** Embase, MEDLINE, PsycINFO, Applied Social Science Index and Abstracts, Cumulative Index to Nursing and Allied Health and grey literature were searched.

**Methods** Eligible studies explored the impact of closure of an OU or the effect of travel distance/time on prespecified maternal or neonatal outcomes. Only studies of women giving birth in high-income countries with universal health coverage of maternity services comparable to the UK were included. Identification of studies, extraction of data and risk of bias assessment were undertaken by at least two reviewers independently. The risk of bias checklist was based on the Cochrane Effective Practice and Organisation of Care criteria and the Newcastle-Ottawa scale. Heterogeneity across studies precluded meta-analysis and synthesis was narrative, with key findings tabulated.

**Results** 31 studies met the inclusion criteria. There was some evidence to suggest an increase in babies born before arrival following OU closures and/or associated with longer travel distances or time. This may be associated with an increased risk of perinatal or neonatal mortality, but this finding was not consistent across studies. Evidence on other maternal and neonatal outcomes was limited but did not suggest worse outcomes after closures or with longer travel times/distances. Interpretation of findings for some studies was hampered by concerns around how accurately exposures were measured, and/or a lack of adjustment for confounders or temporal changes.

**Conclusion** It is not possible to conclude from this review whether OU closure, increased travel distances or times are associated with worse outcomes for the mother or the baby.

**PROSPERO registration number** CRD42017078503.

## Strengths and limitations of this study

► This review is the first to synthesise systematically the current evidence on the impact of closure of obstetric units and of travel distance and travel time to obstetric units on neonatal and maternal outcomes.
► Rigorous systematic review methodology was applied including a sensitive search strategy to ensure all relevant evidence was identified.
► Heterogeneity across included studies precluded any form of meta-analysis.
► A paucity of evidence on a number of outcomes, and methodological concerns for some studies limited conclusions that could be drawn.

## BACKGROUND

Closure of small obstetric units (OUs) and centralisation of obstetric services in larger units has been proposed to increase levels of consultant obstetrician cover to improve safety and limit costs. However, closure of OUs or conversion of OUs to midwifery-led units/community-based services potentially leads to an increase in travel distance or time for women in labour from their home to the nearest OU. Increases in travel time could potentially increase the risk of adverse birth outcomes.

Travel time and distance are widely used as measures to explore the geographical accessibility of health services.[1] In a systematic review,[2] the association between travelling further to healthcare facilities and having worse health outcomes was established, but the review did not include studies of maternity care. The impact of OU closure and increase in travel time/distance to the OU on perinatal and maternal outcomes have not been systematically assessed. One

review[3] evaluating the effects of regionalisation of perinatal services has been published. This concluded that regionalisation programmes appeared to be correlated with improvements in perinatal outcomes but that the evidence was weak. A narrative review[4] included 10 studies that explored travel time and distance to and between maternity services and adverse birth outcomes to inform the consultation on maternity services in Wales. The review was limited to studies reported in English and there was no clear association between travel distance or time and adverse birth outcomes

Therefore, uncertainty remains about the association between OU closure, prolonged time or distance to OUs and adverse perinatal outcomes. Specifically, there is a rise in the risk of babies born before arrival (BBA, also referred to as unplanned out of hospital births). Being BBA is more common before term and has been reported to be associated with higher perinatal mortality (PM).[5] Conversely, Lasswellet *et al*[6] found neonatal mortality (NM) was reduced when services were configured to ensure very preterm infants are born in a large maternity hospital with neonatal intensive care unit (level III NICU). In addition to mortality, Apgar scores (a standardised measure of the physical condition of a newborn infant) and neonatal admission to intensive care provide an indication of perinatal infant health.

The impact on maternal outcomes is also unclear. There are concerns that low-risk women who give birth in larger hospitals may experience more interventions, for example, increased frequency of caesarean section (CS).[7] Along with CS, evidence on maternal mortality (MM) and maternal birth complications such as postpartum haemorrhage (PPH) and maternal blood transfusion, was also sought in this review to identify the potential impact of OU closure on maternal outcomes.

In this review, we aimed to systematically identify, critically appraise and synthesise the evidence relating to: (1) The effect of OU closures on maternal and neonatal outcomes (compared with the surrounding area or a comparable population) and (2) The association between travel distance or time to an OU and maternal and neonatal outcomes.

## REVIEW METHOD

The Meta-Analyses and Systematic Reviews of Observational Studies in Epidemiology (MOOSE) reporting guideline was followed.[8]

### Criteria for considering studies for this review
#### Types of studies
Any quantitative study design with a comparison group was eligible for inclusion. Studies were included from 1990 onwards. The year 1990 was chosen as a cut-off date because significant advances were made in neonatal care in the early 1990s, such as surfactant therapy, assisted ventilation, prophylactic infection control and antenatal steroid therapy, which impacted on the delivery

of maternity services.[9] The quantitative components of mixed methods studies were also eligible. Studies were included if they:

► Explored the impact of closure of an OU on maternal or neonatal outcomes either in a before-and-after comparison (same population catchment area), or a geographical comparison of different areas (comparable populations).

And/or

► Compared maternal and neonatal outcomes after an OU closure and retention or creation of midwifery led units to replace the OU.

► Explored the effect of travel time and/or distance on maternal and neonatal outcomes providing at least two travel times and/or distances from women's homes to the nearest OU.

► Explored maternal and neonatal outcomes following maternal transfer from planned or unplanned home birth to the nearest maternity centre.

We included studies of women giving birth in high-income, the Organisation for Economic Co-operation and development (OECD) countries with universal health coverage (UHC) of maternity services comparable to the UK. The list of OECD countries is shown in online supplemental appendix 1. UHC is defined as healthcare that meets everyone's right to access high quality essential health services where and when they need them without financial difficulty.[10]

### Types of exposures
OU closure: the closure of an OU was compared with no closure of an OU for the same or comparable geographical catchment areas prior to the closure. For a study comparing different geographical areas affected by the closure of an OU, the least affected area was used as a control group. For the purpose of this review, we used the definition of an OU used in the Birthplace Research programme in England,[11] which defined an OU as 'a clinical location in which care is provided by a team, with obstetricians taking primary professional responsibility for women at high risk of complications during labour and birth. Midwives offer care to all women in an OU, whether or not they are considered at high or low risk, and take primary responsibility for women with straightforward pregnancies during labour and birth. Diagnostic and treatment medical services including obstetric, neonatal and anaesthetic care are available on site, 24 hours a day'[11] (P12).

Travel distance or time to the nearest OU: a shorter travel distance or time was compared with a longer travel distance or time. We used the definition of a shorter or a longer time or distance as defined by the included studies. When a study compared several different travel times or distances to the nearest OU, those with the shortest travel distance or time were used as the control group.

The following types of studies were excluded:

► Studies comparing maternal and or neonatal outcomes based on hospital size, level of NICU, type

of hospital or model of care (eg, caseload midwifery care vs consultant care).
► Studies on regionalisation of neonatal care (number of centres with NICUs).
► Studies where a proximity rather than the actual travel time or travel distance was given (eg, rural vs urban, remote vs very remote areas).
► Studies which did not report at least one of the outcomes.

### Review outcomes
The following outcomes were predefined in the study protocol:

### Maternal outcomes
Maternal mortality (MM), caesarean section (CS) (overall, emergency or intrapartum), severe perineal trauma (including third and fourth degree tears), postpartum haemorrhage (PPH), maternal admission to intensive care units (ICU) and maternal blood transfusion.

### Neonatal outcomes
Stillbirth (SB) (overall or intrapartum), neonatal mortality (NM), PM, infant mortality (IM), babies BBA, neonatal unit admission (NNU), Apgar score and hypoxic-ischaemic encephalopathy (HIE).

## REVIEW METHODS
A comprehensive search strategy was developed in collaboration with an information specialist (NR). We searched Embase, Medline, PsycINFO, Applied Social Science Index and Abstracts and Cumulative Index to Nursing and Allied Health databases (from 1990 to February 2019). We also searched the grey literature in the databanks of British Library EThOS, Open Grey and ProQuest Dissertations & Theses Global. National Health Service (NHS) Trusts and Health Boards in the UK were also contacted where we had been able to identify an OU closure to request information about any evaluations that were conducted. The references of eligible studies and relevant reviews were checked to identify additional studies not retrieved by the search. Searches were based on index terms and text words relating to the population/setting (eg, maternity service, pregnancy, neonatal) and exposures (eg, travel/distance or closure/regionalisation). Due to the variable nature of terms and indexing used, the strategy was kept broad by using a range of alternate terms and not limiting by outcome. No language restriction was applied. A sample search strategy for MEDLINE is shown in online supplemental appendix 2.

At least two reviewers (RSM, CT, AP, FA and JH) independently screened the references for relevance against the review eligibility criteria using Eppi-reviewer software (V.4).[12] Full-text study screening was also performed by at least two reviewers (RSM, CT, CO, JH and FA). Disagreements regarding study eligibility were resolved through discussion and consensus within the review team. We contacted authors of relevant studies published as abstracts for further information. Data extraction and risk of bias assessment were undertaken by at least two reviewers (RSM, CT, CO, JH, FA and JD). The risk of bias checklist was adapted from the Effective Practice and Organisation of Care (EPOC)[13] and the Newcastle-Ottawa scale (for case–control studies).[14] Risk of bias assessment included selection of study groups, measurement of exposure and outcomes, missing data and appropriateness of analysis (eg, logistic regression analysis). For case–control studies, selection and comparability of cases and controls were also considered. The review team rated the quality of evidence for each domain in the tool as low, high or unclear risk of bias, or yes, no and unclear in meeting quality criteria.

Results were synthesised narratively and the key findings tabulated. The included studies varied in their study design, categories of exposure, outcomes reported, whether adjusted or unadjusted results were presented and factors adjusted for. This clinical and methodological heterogeneity across the included studies precluded any form of meta-analysis. Prespecified subgroups were risk status of woman (low vs high), parity, gestational age, UK studies compared with non-UK studies and planned versus unplanned CS; formal subgroup analyses were, however, not possible. Evidence regarding OU closure, travel distance and travel time is reported separately, and by outcome. We have highlighted where crude (unadjusted) ORs (cOR) and adjusted ORs (adjOR) have been reported.

## PATIENT AND PUBLIC INVOLVEMENT
We involved our parent, patient and public involvement (PPPI) Stakeholders Network, to explore which outcomes were important from a maternal perspective. The dissemination of findings to stakeholders will be through plain language summaries developed with members of our PPPI stakeholder network.

### Search results
Searches of bibliographic databases and other sources from 1990 to February 2019 yielded 13 271 unique references and the steps of study selection are presented in the Preferred Reporting Items for Systematic Reviews and Meta-Analyses flow chart (figure 1). The eligibility of 295 full-text articles were assessed independently. Two hundred and sixty articles were excluded for various reasons, including: studies conducted in low-income/middle-income countries, comparing different models or levels of maternity care, assessing women's transfer from primary to secondary maternity centres, or not providing quantifiable measures of travel time/distance (full list available from authors). Thirty-one studies, reported in 35 articles, met the review eligibility criteria (figure 1). One study[5] included information on both OU closure and travel distance. Ten studies provided information on OU closures, 7 studies compared different travel distances

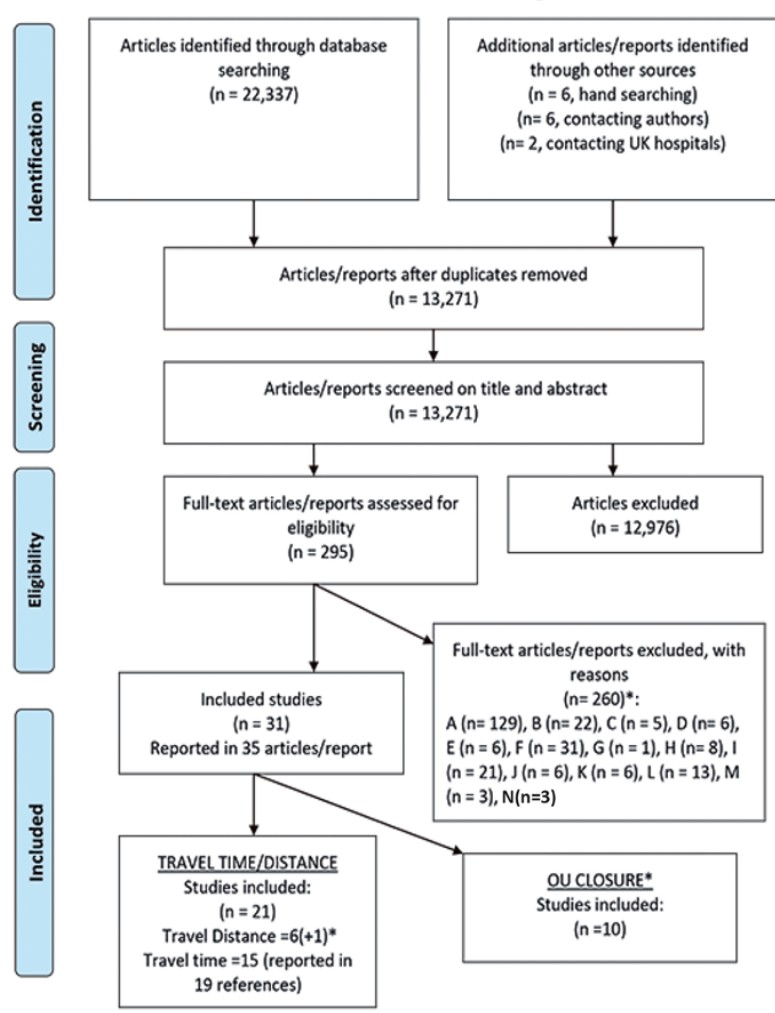

**Figure 1**   PRISMA flow diagram. PRISMA, Preferred Reporting Items for Systematic Reviews and Meta-Analyses.

from women's homes to the nearest OU and 15 studies compared different travel times from women's homes to the nearest OU.

## REVIEW RESULTS
### Evidence from OU closures
A detailed description of the ten included OU closure

studies is presented in table 1. Three studies were from the UK, with two reported as abstracts only[15 16] and one an unpublished data series from East Lancashire Hospitals NHS Trust, UK (East Lancashire Hospitals NHS Trust, unpublished data 2017). There appeared to be overlap between populations reported in two studies (East Lancashire Hospitals NHS Trust, unpublished data 2017) and Fleming et al.[15] Three studies were from Scandinavia,[7 17 18] three from Canada[19–21] and one from France.[5]

Seven studies compared adverse birth outcomes before and after centralisation of services, which included closure of varying numbers of OUs. All three studies from the UK[15 16] and (East Lancashire Hospitals NHS Trust, unpublished data 2017), examined the impact of the amalgamation of two OUs. Four studies were published after 2014[15–17 20]; the earliest was from 1990.[21] Three studies included all births[17 18 21]; the other studies varied in their eligibility criteria, for example, restricting the analysis to singletons pregnancies, live births, various gestational ages and birth weight, hospital births or location. Reporting of eligibility criteria and participant characteristics across studies was inconsistent (table 1).

### Risk of bias assessment

Risks of bias related to a lack of reporting of whether changes over time (other than closure/reconfiguration) could have influenced the findings, with only two[17 20] of 10 studies reporting that temporal variation was adjusted for in the analysis (table 2). Further, 5 out of 10 studies either did not adjust results for potential confounding factors or provided insufficient information to know whether this was undertaken. Five out of 10 studies did not provide sufficient information to gauge the completeness of data. Half of the studies reported and used appropriate data analysis methods. Other potential sources of bias (eg, relating to selection, exposure and outcome) were less of a concern due to the use of routinely collected registry data before and after the closure and the objective nature of most outcomes.

### Findings

A summary of maternal and neonatal outcomes is presented in table 3.

#### *Maternal outcomes*
##### Maternal mortality (MM)
In the two studies that reported MM,[16 20] the number of deaths (<5) was too low to allow comparisons between the preclosure and postclosure groups.

#### Caesarean section (CS) (overall or intrapartum)
Total CS rates were reported in four studies.[7 15 20 21] One UK study[15] reported a decline in CS rates following the amalgamation of two units from 26.1% to 21.5%.

A Norwegian study,[7] reported an increase in CS rates from 13.1% to 16.4% following OU closure, (cOR 1.31, 95% CI 1.27 to 1.35) as did two Canadian studies[21] (cOR

1.13, 95% CI 1.09 to 1.18) and (cOR 1.10, 95% CI 1.01 to 1.19).[20] No adjusted results were reported.

#### Emergency CS
Emergency CS rates were reported in one UK study,[16] which found no difference before/after the amalgamation of two OUs (cOR 0.95, 95% CI 0.86 to 1.05).

#### *Severe perineal trauma (third or fourth degree tear)*
Two studies[16 20] reported this outcome and found no statistically significant difference between the before/after closure groups. The incidence of the outcome in both studies was low (<3%).

PPH—No studies reported this outcome.

#### Maternal admission to ICU
Two studies[16 20] found no significant difference before/after the amalgamation of two OUs in the number of women requiring admission to ICU (cOR 0.80, 95% CI 0.44 to 1.46).[16] The numbers in one study[20] were too small (<5) to allow a comparison.

#### Maternal blood transfusion
One study[20] found no significant differences before/after OU closure (cOR 0.82, 95% CI 0.55 to 1.21). The incidence of the outcome was low (<1% of women).

### Neonatal outcomes

#### Stillbirth (SB) (overall or intrapartum)
Three studies examined the impact of OU closure on SB. One unpublished UK study (East Lancashire Hospitals NHS Trust, unpublished data 2017) showed a statistically significant reduction in SB over the period after the amalgamation of two units (cOR 0.61, 95% CI 0.47 to 0.78). Similar findings were seen in one study from Canada[22] during post closure years (cOR 0.80, 95% CI 0.67 to 0.96). A third study from the UK[16] found no difference in SB rates after OU closure.

#### Neonatal mortality (NM)
Three studies reported this outcome. Two studies from the UK[16] and (East Lancashire Hospitals NHS Trust, unpublished data 2017) showed no statistically significant difference in the rate of NM in the years after OU closure (cOR 1.33, 95% CI 0.81 to 2.17; cOR 0.80, 95% CI 0.29 to 2.26). A study from Norway[17] also reported no difference (no OR presented).

#### Perinatal mortality (PM)
Two studies reported this outcome. In a study from Norway,[7] PM was significantly lower following OU closure (cOR 0.76, 95% CI 0.58 to 0.98). A Canadian study[22] also reported a significant reduction in PM after OU closure (cOR 0.75, 95% CI 0.64 to 0.87).

#### Infant mortality (IM)
One study[17] reported this outcome, IM rates were 'not statistically elevated' after the closure of thirteen hospitals in Norway.

**Table 1** Description of included studies—OU closures

| Author, year, country | Study design and setting | Study objectives | Study period | Eligibility criteria | Participant characteristics | Description of exposure (change over time) | Services context information | Review outcomes | |
|---|---|---|---|---|---|---|---|---|---|
| **Perinatal** | | | | | | | | | |
| **Maternal** | | | | | | | | | |
| **UK studies** | | | | | | | | | |
| **Fleming/East Lancashire study** | | | | | | | | | |
| Fleming[15] 2013, UK (abstract) | Retrospective analysis pre and post service reconfiguration East Lancashire maternity services | To examine the effect of major service reconfiguration on CS rates | Time of reconfiguration: November 2010 Time of analysis: January–June 2010 vs January–June 2012 | NR | N: NR Characteristics: NR | Service reconfiguration Pre-change: 2 OUs Post-change: 1 OU +3 MW-led units | Universal state provision of maternity care. Approx 7000 births/yr at the new unit. | NR | CS |
| East Lancashire Hospitals NHS Trust 2017, UK (unpublished-data) | Retrospective population-based analysis of routinely collected data of service reconfiguration East Lancashire, Blackburn and Burnley | To review outcomes after centralisation of services on the Burnley general hospital site | Time of reconfiguration: November 2010 Time of analysis: 2009–2017 | NR | 2009–2017 n=53 870 births (2010 data excluded) Characteristics: NR | Amalgamation of 2 OUs Prechange: 2 OUs Postchange: 1 OU plus two birth centres | Universal state provision of maternity care. Changes driven by pressure on staff rotas, European Working Time Directive, and desire to maintain high quality service. 6000–7000 births/yr at new unit. | BBA; SB; NM | NR |
| Mackie et al[16], 2014, UK (abstract & (unpublished data) | Retrospective analysis pre- and post- service reconfiguration Pennine Acute Trust: Royal Oldham Hospital, North Manchester General Hospital and Fairfield General Hospital | To assess the effect of the amalgamation of 2 OUs to form a 'super-centre' with increased consultant labour ward cover | Time of OUs amalgamation 2011 Time of analysis: prechanges 2010–2011 vs postchanges 2011–2013 | NR | Preamalgamation n=5422 Postamalgamation n=5046 Characteristics: NR | Service reconfiguration Prechange: 2 OUs Postchange: 1 OU | Universal state provision of maternity care. Approx 5000 births/year at the new unit. | SB; NM; BBA | Maternal mortality; Emergency CS; third and fourth degree perineal tea; Maternal transfer to ICU |
| **Other European Studies** | | | | | | | | | |

Continued

**Table 1** Continued

| Author, year, country | Study design and setting | Study objectives | Study period | Eligibility criteria | Participant characteristics | Description of exposure (change over time) | Services context information | Review outcomes |
|---|---|---|---|---|---|---|---|---|
| **Perinatal** | **Maternal** | | | | | | | |
| Blondel et al[5], 2011, France* | Retrospective population-based analysis of routinely collected data, OU closure Across France | To report on BBA incidence in relation to distance from OU and the closure impact on different sociodemographic groups | Time of OU closure: 2003 and 2006 Time of analysis: 2005–2006 | Included: Singleton births Excluded: Municipalities if >8% missing data, or high OOH rates. Departments excl. if >20% births already excl. | n=1 349 751 births; OOH n=5740 N Births 1349 to 751 Age (yrs) (n): <20–26 152 20–24 - 188 350 25–29 - 427 462 30–34 - 442 089 35–39 - 213 534 40+ - 52 164 Nullip (n) 774 460 SES: occupation professional (n) 217 045 intellectual 325 746 admin 266 000 retail 122 727 skilled 49 201 unskilled 84 664 none 184 368 Ethnicity, education: NR | Closure of maternity unit Pre-change: no of OUs NR Postchange: Closure of units within 15 km radius of home, number of units closed NR | Centralisation of births in larger units due to safety concerns, financial pressure, efficiency savings, and staff shortage | BBA | NR |
| Hemminki et al[18], 2011, Finland | Retrospective population-based analysis of routinely collected data, OU closure Across Finland and a specific district Uusimaa | To describe centralisation trend, unplanned out of hospital births, perinatal mortality (PM), health and birth outcomes in areas served by different levels hospitals | Time of OU closure: 1991–2008 Time of data analysis: Finland 1991–2008; Uusimaa district 2004–2008 | Inclusion: All births Exclusion: NR | 1991–2008 n=474 419 Characteristics: NR | Centralisation of births, maternity units no declined Pre-change: 49 OUs in 1991 Post-change: 34 OUs in 2008 | Universal access to maternity care, minimal private care. Pre- and postnatal care decentralised, birth hospital-based service, care of high-risk pregnancies centralised. Mean no births/hospital increased from 1339 to 1733 over study period. | BBA | NR |

Continued

**Table 1** Continued

| Author, year, country | Study design and setting | Study objectives | Study period | Eligibility criteria | Participant characteristics | Description of exposure (change over time) | Services context information | Review outcomes |
|---|---|---|---|---|---|---|---|---|
| **Perinatal** | | | | | | | | |
| **Maternal** | | | | | | | | |
| Engjom et al[7], 2014, Norway | Retrospective population-based study, 3 cohort and two cross-sectional studies, OU closure Across Norway | To assess the availability of OUs, the risk of unplanned delivery outside OU and maternal morbidity | Cohort: 1979–2009 Cross-sectional: 2000 and 2010 Time of OUs closure: 1979–2009 Time of analysis: Cohort: 1979–2009 Cross-sectional: 2000 and 2010 | Included: Age 15–49 years, known place of birth, GA ≥22 wks and/or bthwt >500 g Excluded: Missing maternal address, planned home birth | 1979–1983 n=252 621 2004–2009 n=409 432 Characteristics: NR | Declined in no of OUs in Norway Prechange: 95 OUs in 1979 Post-change: 51 OUs in 2009 | Universal access to maternity care; relatively dispersed population | PM; CS |
| Grytten et al[17], 2014, Norway | Retrospective population-based analysis of routinely collected data, OU closure Across Norway | To study whether neonatal and infant mortality (IM) were independent of the type of hospital in which the delivery was carried out | Time of closures: between 1979 and 2005 Time period for analysis: 5 years pre and postclosure for each hospital | Inclusion: All births Exclusion: NR | n=33 677 Characteristics: NR | Centralisation /OU closures No of local hospitals fell from 43 to 26 between 1979 and 2005. 17 maternity wards in local hospitals closed Prechange: 22 hospitals with neonatal department; 43 local hospitals; 30 maternity clinics Postchange: 22 hospitals with neonatal department; 26 local hospitals, 10 maternity clinics | Universal free access to maternity care. Low-risk births in local hospitals, high risk in central/ regional hospitals. Births in central/ regional hospitals increased from 65%–81% over study period. | NM; IM; NR |
| **Canadian Studies** | | | | | | | | |
| Le Coutou et al[21], 1990, Canada | Retrospective population-based analysis of routinely collected data, OU closure Montreal metropolitan area | To describe the evolution of obstetric practice in Montreal metropolitan area before/ after closure of units | Time of closure: 1984–1985 Time of analysis: 1981–1985 | Inclusion: All births Exclusion: NR | 1981–1984 n=1 28 688 Characteristics: NR | 7/13 university hospitals, 5/13 specialist hospitals, 4/13 smaller units closed Prechange: 39 units in 1981 Postchange:16 | State provision of maternity care. Closures due to budgetary restrictions. | NR; Overall CS |

Continued

**Table 1** Continued

| Author, year, country | Study design and setting | Study objectives | Study period | Eligibility criteria | Participant characteristics | Description of exposure (change over time) | Services context information | Review outcomes |
|---|---|---|---|---|---|---|---|---|
| **Perinatal** | | | | | | | | |
| **Maternal** | | | | | | | | |
| Allen et al,[22] 2004, Canada | Retrospective population-based analysis of routinely collected data, OU closure Eastern, Northern, Western, and Central in Nova Scotia | To evaluate the effect of hospital closures on critical obstetrical interventions and perinatal outcomes in rural communities | Preclosure: 1988–1993 Post-closure: 1996–2002 Time of analysis: 1988–1993 vs 1996–2002 | Inclusion: All births Exclusion: Delivery <20 weeks; bthwt <500 g; triplets+; major congenital anomaly | 1988–93 n=69 213 1996–2002 n=63 510 Range % Age >34 yrs: 5.6–14.8; Nullip 39.4–46.8 Twins 1.0–1.3 Ethnicity, socioeconomic status, education: NR | 1988–1993 =27 hospitals 1996–2002 =19 hospitals Reduction in maternity units from 42 to 11 between 1970 and 2002 Pre-change: 42 units in 1970 Post-change: 11 units in 2002 | State provision of maternity care. Reduction in no of units and physicians due to financial constraints and difficulty maintaining clinical competence and confidence. | SB; Foetal/ neonatal mortality (NM) | NR |
| Hutcheon et al[20], 2017, Canada | Retrospective population based analysis of routinely collected data, OU closure 25 communities within British Columbia, Canada | To examine the effect of obstetric service closures on intrapartum outcomes | 1998–2014 Time of closures: between 2000 and 2012 Time of analysis: 1998–2014 | Inclusion: All births recorded in British Columbia Perinatal Data Registry (99% of deliveries) Exclusion: Communities close to larger metropolitan areas and or uncertainty about dates of service closures. | Pre-closure n=5796 Median maternal age 27 years (IQR 23–31); Nullip 39.3% Post-closure n=6153 Median maternal age 28 years (IQR 24–32); Nullip 40.7% Ethnicity, Socioeconomic status, education: NR | Centralisation /OU closures Between 1998 and 2014 one-third of hospitals stopped providing maternity services Pre-change: 21 hospitals with obstetric services Postchange: Obstetric services closed in same 21 hospitals | State provision of maternity care. Centralisation of obstetric services, majority of hospital closures in low-volume hospitals | BBA; perinatal/ NM; NNU admission | Overall CS; Maternal mortality; third/4th degree perineal tear, blood transfusion, maternal admission to ICU |

*Blondel et al[5] is also included in travel distance.

Approx, approximately; BBA, Born before arrival; bthwt, birth weight; CS, caesarean section; excl, excluded; GA, gestational age; ICU, intensive care unit; MW, midwife; NHS, National Health Services; NNU, neonatal unit; NR, not reported; Nullip, nulliparous; OU, obstetric unit; SB, stillbirth; SES, socioeconomic status; wo, without; Yr, year.

**Table 2** Risk of bias—obstetric unit (OU) closure studies

| Author, year, country | Study sample selection bias | Bias in measurement of exposure | Bias in measurement of outcomes | Attrition bias | Analysis method reported and appropriate | Closure independent of other changes over time | Potential confounders adjusted for and listed |
|---|---|---|---|---|---|---|---|
| **UK Studies** | | | | | | | |
| *Fleming/East Lancashire study* | | | | | | | |
| Fleming[15], 2013, UK (abstract) | LOW All births in East Lancashire Maternity Services catchment | LOW All births in catchment area affected by the closure No of OUs closed reported | LOW Objective outcome (CS) | UNCLEAR Not reported | UNCLEAR None reported | UNCLEAR None reported | UNCLEAR None reported |
| East Lancashire Hospitals NHS Trust 2017, UK (unpublished data) | UNCLEAR Unpublished data, no details | UNCLEAR Unpublished data, no details | LOW Objective outcomes (BBA, SB, NM) | UNCLEAR Unpublished data, no details | UNCLEAR Unpublished data, no details | UNCLEAR Unpublished data, no details | UNCLEAR Unpublished data, no details |
| Mackie et al[16], 2014, UK (abstract & unpublished) | LOW Data from Maternity Information System | LOW All births in catchment area affected by the closure. No of OUs closed reported | LOW Objective outcomes (SB, NM, BBA, MM, ICU admission, perineal tears) | UNCLEAR Not reported | UNCLEAR None reported | UNCLEAR None reported | UNCLEAR None reported |
| **Other European Studies** | | | | | | | |
| Blondel et al[5], 2000, France* | LOW Data from birth certificates | LOW No of OUs closed reported | LOW Objective outcome (BBA) | LOW 11% excluded due to missing data | LOW Analysis method was described and appropriate, a multi-level model analysis | UNCLEAR None reported | LOW Adjusted for maternal age, occupational category and rurality |
| Hemminki et al[18], 2011, m Finland | LOW Data from Finnish medical birth register | LOW All births in catchment area affected by the closure No of OUs closed reported | LOW Objective outcome (BBA) | LOW Births with missing information excluded (<0.05%) | LOW Analysis method was described and appropriate, a regression model with adjusted analysis | UNCLEAR None reported | LOW Adjusted for Parity, plurality, age, socioeconomic status and smoking |
| Engjom et al[7], 2014, Norway | LOW Data from Medical Birth Registry of Norway | LOW All births in Norway affected by the closure No of OUs closed reported | LOW Objective outcome (BBA) | LOW All units report to Medical Birth Registry | LOW Analysis method appropriate, a logistic regression model, crude and adjusted results given | UNCLEAR None reported | LOW Adjusted for maternal age, parity, education and partner status |

Continued

**Table 2** Continued

| Author, year, country | Study sample selection bias | Bias in measurement of exposure | Bias in measurement of outcomes | Attrition bias | Analysis method reported and appropriate | Closure independent of other changes over time | Potential confounders adjusted for and listed |
|---|---|---|---|---|---|---|---|
| Grytten et al[17], 2014, Norway | LOW Data from Medical Birth Registry of Norway | LOW All births in Norway No of OUs closed reported | LOW Objective outcomes (NM, Infant mortality) | LOW All maternity units report to Medical Birth Registry | Unclear Difference-in-difference statistical method used, but reporting of findings were unclear | LOW Adjusted for trend in infant outcomes based on local hospitals that were not closed. | LOW Maternal age, immigrant status, level of education, marital status, predisposing medical factors and characteristics of the birth |
| **Canadian Studies** | | | | | | | |
| Le Coutour et al[21], 1990, Canada | LOW Data from MED-ECHO – regional data collection system | LOW All births in catchment area No of OUs closed reported | LOW Objective outcome (CS) | UNCLEAR No information | UNCLEAR No details on data analysed method | UNCLEAR None reported | HIGH No adjustment |
| Allen et al,[22] 2004, Canada | LOW Data from Nova Scotia Atlee Perinatal Database | LOW All births in catchment area affected by the closure No of OUs closed reported | LOW Objective outcomes (SB, NM) | UNCLEAR Population based dataset but no information about missing data | LOW Analysis method appropriate and data from logistic regression models were reported | UNCLEAR None reported | LOW Maternal age, smoking and maternal diseases |
| Hutcheon et al[20], 2017, Canada | LOW Data from British Columbia Perinatal registry | LOW All births in catchment area affect by the closure No of OUs closed reported | LOW Except for third/4th degree tears. Objective outcomes (BBA, PM, NM, ICU admission, CS, MM, blood transfusion, Maternal admission to ICU) | LOW >99% complete | LOW Used a within-community fixed-effects design and Poisson regression | LOW Using difference in difference analysis which separates the effect of the closure from underlying time trends of reported outcomes | HIGH No adjustment |

*Blondel et al 2011 included in travel distance and OU closure.

BBA, born before arrival; CS, caesarean section; ICU, intensive care unit; MM, maternal mortality; NHS, National Health Service; NM, neonatal mortality; PM, perinatal mortality; SB, stillbirth.

**Table 3** Outcomes—obstetric unit (OU) closure

| Outcomes | Author, Year, Country | Exposure and comparator groups | Participants (N, n, %) | Findings | | | | |
|---|---|---|---|---|---|---|---|---|
| **MATERNAL OUTCOMES** | | | | | | | | |
| Maternal mortality (MM) | Mackie et al[16], 2014, UK (unpublished data) | Before and after amalgamation of 2 OUs | Year | Year | Deliveries (n=15349) | MM n (%) | Crude OR (95% CI) | Adjusted OR (95% CI) |
| | | | Pre 2010–2011 | Pre 2010–2011 | 5354 | 1 (0.02) | 1 | NR |
| | | | Post 2011–2013 | Post 2011–2013 | 9995 | 1 (0.01) | 0.54 (0.03 to 8.56) | NR |
| | Hutcheon et al[20], 2017, British Columbia, Canada | Before and after OU closure (1998–2014) | Closure status | Closure status | Deliveries (n=11949) | Maternal deaths n (%) | Crude OR (95% CI) | Adjusted OR (95% CI) |
| | | | Preclosure | Preclosure | 5796 | <5 (<0.09) | NR | NR |
| | | | Postclosure | Postclosure | 6153 | <5 (<0.08) | NR | NR |
| | | No significant difference pre/post closure in adverse events during labour and delivery | | | | | | |
| Caesarean section (CS) (overall or intrapartum) | Fleming et al[15], 2013, UK (abstract) | Before and after closure of OU in 2010 | Closure status | Closure status | Deliveries (n=NR) | CS n (%) | Crude OR (95% CI) | Adjusted OR (95% CI) |
| | | | Preclosure, early 2010 | Preclosure, early 2010 | NR | (NR) 26.1 | NR | NR |
| | | | Postclosure, 2012 | Postclosure, 2012 | NR | (NR) 21.5 | NR | NR |
| | | | Proportions of CS presented with no other data. | | | | | |
| | Engjom et al[7], 2014, Norway | 2000 compared with 2009 during which time number of OUs declined from 47 to 41 | Year | Year | Deliveries (n=2,177,934) | CS n (%) | Crude OR (95% CI) | Adjusted OR (95% CI) |
| | | | Pre 2000 | Pre 2000 | 58632 | 7653 (13.10) | 1 | NR |
| | | | Post 2009 | Post 2009 | 61895 | 10154 (16.41) | 1.31 (1.27 to 1.35) | NR |
| | Le Coutour et al[21], 1990, Quebec, Canada | Before and after closure of OUs between 1982 and 1983 | Year | Year | Deliveries (n=64274) | CS n (%) | Crude OR (95% CI) | Adjusted OR (95% CI) |
| | | | Pre 1981 | Pre 1981 | 32807 | 5852 (17.84) | 1 | NR |
| | | | Post 1984 | Post 1984 | 31467 | 6214 (19.7) | 1.13 (1.09 to 1.18) | NR |
| | Hutcheon et al[20], 2017, British Columbia, Canada | Before and after OUs, closure (1998–2014) | Closure status | Closure status | Deliveries (n=11949) | CS n (%) | Crude OR (95% CI) | Adjusted OR (95% CI) |
| | | | Preclosure | Preclosure | 5796 | 1387 (23.93) | 1 | NR |
| | | | Postclosure | Postclosure | 6153 | 1579 (25.70) | 1.10 (1.01 to 1.19) | NR |
| Emergency caesarean section (CS) | Mackie et al[16], 2014, UK (abstract only) | Before and after amalgamation of two OUs | Year | Year | Deliveries (n=15349) | Emergency CS n (%) | Crude OR (95% CI) | Adjusted OR (95% CI) |
| | | | Pre 2010–2011 | Pre 2010–2011 | 5354 | 739 (13.80) | 1 | NR |
| | | | Post 2011–2013 | Post 2011–2013 | 9995 | 1322 (13.23) | 0.95 (0.86 to 1.05) | NR |
| Severe perineal trauma (third or fourth degree tear) | Mackie et al[16], 2014, UK (unpublished data) | Before and after amalgamation of two OUs | Year | Year | Deliveries (n=15349) | 3rd & fourth n (%) | Crude OR (95% CI) | Adjusted OR (95% CI) |
| | | | Pre 2010–2011 | Pre 2010–2011 | 5354 | 133 (2.48) | 1 | NR |
| | | | Post 2011–2013 | Post 2011–2013 | 9995 | 276 (2.76) | 1.11 (0.90 to 1.37) | NR |
| | Hutcheon et al[20], 2017, British Columbia, Canada | Before and after OU closure (1998–2014) | Closure status | Closure status | Deliveries (n=11949) | third or fourth degree tear n (%) | Crude OR (95% CI) | Adjusted OR (95% CI) |
| | | | Preclosure | Preclosure | 5796 | 136 (2.40) | 1 | NR |
| | | | Postclosure | Postclosure | 6153 | 174 (2.82) | 1.21 (0.96 to 1.52) | NR |
| Postpartum haemorrhage | No studies | | | | | | | |

Continued

**Table 3** Continued

| Outcomes | Author, Year, Country | Exposure and comparator groups | Participants (N, n, %) | | | Findings | Crude OR (95% CI) | Adjusted OR (95% CI) |
|---|---|---|---|---|---|---|---|---|
| Maternal admission to ICU | Mackie et al[16], 2014, UK (abstract only) | Before and after amalgamation of two OUs | Year | Deliveries (n=15349) | n (%) | Year | Crude OR (95% CI) | Adjusted OR (95% CI) |
| | | | Pre 2010–2011 | 5354 | 18 (0.34) | Pre 2010–2011 | 1 | NR |
| | | | Post 2011–2013 | 9995 | 27 (0.27) | Post 2011–2013 | 0.80 (0.44 to 1.46) | NR |
| | Hutcheon et al[20], 2017, British Columbia, Canada | Before and after OU closure (1998–2014) | Closure status | Deliveries (n=11949) | ICU admission n | Closure status | Crude OR (95% CI) | Adjusted OR (95% CI) |
| | | | Preclosure | 5796 | <5 | Preclosure | NR | NR |
| | | | Postclosure | 6153 | <5 | Postclosure | NR | NR |
| Maternal Blood transfusion | Hutcheon et al[20], 2017, British Columbia, Canada | Before and after OU closure (1998–2014) | Closure status | Deliveries (n=11949) | Blood transfusion n (%) | Closure status | Crude OR (95% CI) | Adjusted OR (95% CI) |
| | | | Preclosure | 5796 | 53 (0.91) | Preclosure | 1 | NR |
| | | | Postclosure | 6153 | 46 (0.75) | Postclosure | 0.82 (0.55 to 1.21) | NR |
| **NEONATAL OUTCOMES** | | | | | | | | |
| Stillbirth (SB) | East Lancashire Hospitals NHS Trust, 2017, UK (unpublished data) | Before and after amalgamation of two obstetric units (OUs) in 2010 | Year | Deliveries (n=53870) | SB n (%) | Year | Crude OR (95% CI) | Adjusted OR (95% CI) |
| | | | Pre 2009 | 6492 | 75 (1.16) | Pre 2009 | 1 | NR |
| | | | Post 2011–2017 | 47378 | 333 (0.70) | Post 2011–2017 | 0.61 (0.47 to 0.78) | NR |
| | | | Stillbirth>24 weeks | | | | | |
| | Allen et al,[22] 2004, Nova Scotia, Canada | Before and after closure of OUs | Year | Deliveries (n=132723) | SB n (%) | Year | Crude OR (95% CI) | Adjusted OR (95% CI) |
| | | | Pre 1988–93 | 69213 | 291 (0.42) | Pre 1988–1993 | 1 | NR |
| | | | Post 1996–2002 | 63510 | 214 (0.34) | Post 1996–2002 | 0.80 (0.67 to 0.96) | NR |
| | Mackie et al[16], 2014, UK (unpublished data) | Before and after amalgamation of 2 OUs | Year | Deliveries (n=15552) | SB n (%) | Year | Crude OR (95% CI) | Adjusted OR (95% CI) |
| | | | Pre 2010–2011 | 5422 | 29 (0.53) | Pre 2010–2011 | 1 | NR |
| | | | Post 2011–2013 | 10130 | 60 (0.59) | Post 2011–2013 | 1.11 (0.71 to 1.73) | NR |
| Neonatal mortality (NM) | East Lancashire Hospitals NHS Trust, 2017, UK (unpublished data) | Before and after amalgamation of two obstetric units in 2010 | Year | Deliveries (n=53870) | NM n (%) | Year | Crude OR (95% CI) | Adjusted OR (95% CI) |
| | | | Pre 2009 | 6492 | 4 (0.06) | Pre 2009 | 1 | NR |
| | | | Post 2011–2017 | 47378 | 39 (0.08) | Post 2011–2017 | 1.33 (0.81 to 2.17) | NR |
| | | | NM not defined | | | | | |
| | Mackie et al[16], 2014, UK (unpublished data) | Before and after amalgamation of 2 OUs | Year | Deliveries (n=15552) | NM n (%) | Year | Crude OR (95% CI) | Adjusted OR (95% CI) |
| | | | Pre 2010–2011 | 5422 | 6 (0.11) | Pre 2010–2011 | 1 | NR |
| | | | Post 2011–2013 | 10130 | 9 (0.90) | Post 2011–2013 | 0.80 (0.29 to 2.26) | NR |
| | Grytten et al[17], 2014, Norway | Before and after 13 hospital closures | Year | Deliveries (n=33677) | NM n (%) | Year | Crude OR (95% CI) | Adjusted OR(95% CI) |
| | | | 5 years before | 16297 | NR | 5 years before | NR | NR |
| | | | 5 years after | 17380 | NR | 5 years after | NR | NR |
| | | | No statistically significant difference | | | | | |

Continued

**Table 3** Continued

| Outcomes | Author, Year, Country | Exposure and comparator groups | Participants (N, n, %) | | Findings | | | |
|---|---|---|---|---|---|---|---|---|
| Perinatal mortality (PM) | Engjom et al[7], 2014, Norway | 2000 compared with 2009 during which time number of OUs declined from 47 to 41 | Year | | Year | PM n (/1000 births) | Crude OR (95% CI) | Adjusted OR(95% CI) |
| | | | | | Pre 2000 | 124 (2.11) | 1 | NR |
| | | | | | Post 2009 | 99 (1.60) | 0.76 (0.58 to 0.98) | NR |
| | | | Deliveries (n=2,177,934) | PM (Intrapartum & neonatal death<24 hours, both live & stillborn) | | | | |
| | Allen et al[22], 2004, Nova Scotia, Canada | Before and after closure of OUs | Years | | Years | PM n (%) | Crude OR (95% CI) | Adjusted OR (95% CI) |
| | | | | | Pre 1988–93 | 422 (0.61) | 1 | NR |
| | | | | | Post 1966–2002 | 278 (0.43) | 0.75 (0.64 to 0.87) | NR |
| | | | Deliveries (n=132 723) | Pre 1988–1993 | | | | |
| | | | | Post 1996–2002 | | | | |
| | | | | Foetal/neonatal mortality not defined | | | | |
| Infant mortality | Grytten et al[17], 2014, Norway | Before and after 13 hospital closures | Year | | Years | IM n (%) | Crude OR (95% CI) | Adjusted OR (95% CI) |
| | | | | | 5 years before | NR | NR | NR |
| | | | | | 5 years after | NR | NR | NR |
| | | | Deliveries (n=33 677) | 5 years before | | | | |
| | | | 16 297 | 5 years after | | | | |
| | | | 17 380 | No significant difference in infant mortality | | | | |

Continued

 Malouf RS, *et al. BMJ Open* 2020;**10**:e036852. doi:10.1136/bmjopen-2020-036852

**Table 3** Continued

| Outcomes | Author, Year, Country | Exposure and comparator groups | Participants (N, n, %) | | Findings | | | | |
|---|---|---|---|---|---|---|---|---|---|
| Born before arrival (BBA) | East Lancashire Hospitals NHS Trust, 2017, UK (unpublished data) | Before and after amalgamation of two obstetric units | Year | Deliveries (n=53870) | BBA n (%) | Year | | Crude OR (95% CI) | Adjusted OR (95% CI) |
| | | | Pre 2009 | 6492 | 25 (0.39) | Pre 2009 | | 1 | NR |
| | | | Post 2011–2017 | 47378 | 341 (0.72) | Post 2011–2017 | | 1.88 (1.25 to 2.82) | NR |
| | Mackie et al[16], 2014, UK (unpublished data) | Before and after amalgamation of 2 OUs | Year | Deliveries (n=15349) | BBA n (%) | Year | | Crude OR (95% CI) | Adjusted OR (95% CI) |
| | | | Pre 2010–2011 | 5354 | 11 (0.21) | Pre 2010–2011 | | 1 | NR |
| | | | Post 2011–2013 | 9995 | 26 (0.26) | Post 2011–2013 | | 1.28 (0.63 to 2.60) | NR |
| | Blondel et al[6], 2011, France | OU closure within 15km radius 2003–2006 | Yrs 2003–2006 | Deliveries (n=1,349,751) | BBA n (/1000 births) | Yrs 2003–2009 | | Crude OR (95% CI) | Adjusted OR (95% CI) |
| | | | No closure | 1001858 | 4531 (4.52) | No closure | | 1 | 1 |
| | | | Closure within 15km radius | 347893 | 1209 (3.47) | Closure within 15km radius | | 0.77 (0.72 to 0.82) | 0.91 (0.84 to 1.00) |
| | Engjom et al[7], 2014, Norway | 1979–83 compared with 2004–09, number of emergency OUs declined from 47 to 41 | Year | Deliveries (n=662053) | BBA n (%) | Year | | Crude OR (95% CI) | Adjusted OR (95% CI) |
| | | | Pre 1979–83 | 252621 | 984 (0.39) | Pre 1979–83 | | 1 | 1 |
| | | | Post 2004–09 | 409432 | 2832 (0.69) | Post 2004–09 | | 1.8 (1.6 to 1.9) | 2.0 (1.9 to 2.2) |
| | Hemminki et al[18], 2011, Finland | Centralisation of hospitals over years 1991–2008 | Year | Births (N) | Unplanned BBA n(/1000) | Year | Planned or unplanned BBA n(/1000) | Crude OR (95% CI) | NR |
| | | | Pre 1991 | 65632 | – | Pre 1991 | 68 (1.0) | 1 | NR |
| | | | Post 2004–2008 | 56873 | 222 (3.76) | Post 2004–2008 | 243 (4.1) | 4.14 (3.16 to 5.41) | NR |
| | | | Total N (1991–2008)=122505 | | | | | | |
| | Hutcheon et al[20], 2017, British Columbia, Canada | Before and after OU closure (1998–2014) | Closure status | Deliveries (n=11949) | Unplanned BBA n (%) | Closure status | | Crude OR (95% CI) | Adjusted OR (95% CI) |
| | | | Preclosure | 5796 | 30 (0.5) | Preclosure | | 1 | NR |
| | | | Postclosure | 6153 | 109 (1.8) | Postclosure | | 3.47 (2.31 to 5.20) | NR |
| Neonatal unit admission (NNU)>2days or transfer within 24 hours of birth to ICU facility for newborn >/=2500g | Hutcheon et al[20], 2017, British Columbia, Canada | Before and after OU closure (1998–2014) | Closure status | Deliveries (n=11949) | NNU admission n (%) | Closure status | | Crude OR (95% CI) | Adjusted OR (95% CI) |
| | | | Preclosure | 5796 | 68 (1.17) | Preclosure | | 1 | NR |
| | | | Postclosure | 6153 | 28 (0.46) | Postclosure | | 0.39 (0.25 to 0.60) | NR |
| Apgar score (5min Apgar score<7) | Hutcheon et al[20], 2017, British Columbia, Canada | Before and after OU closure (1998–2014) compared with communities unaffected by closure | Closure status | Deliveries (n=11949) | 5min Apgar score<7 n (%) | Closure status | | Crude OR (95% CI) | Adjusted OR (95% CI) |
| | | | Preclosure | 5796 | 71 (1.22) | Preclosure | | 1 | NR |
| | | | Postclosure | 6153 | 85 (1.28) | Postclosure | | 1.13 (0.82 to 1.55) | NR |
| Hypoxic Ischaemic Encephalopathy (HIE) | No studies | | | | | | | | |

ICU, intensive care unit; NR, not reported.

## Born before arrival (BBA)

Six studies reported this outcome, with four suggesting a statistically significant increase in BBA following OU closure. Data from East Lancashire Hospitals NHS Trust (East Lancashire Hospitals NHS Trust, unpublished data 2017) showed the BBA rate almost doubled over the 10-year period (cOR 1.88, 95% CI 1.25 to 2.82). Studies from Norway[7] and Finland[18] also found that the BBA rate increased over a similar period (cOR 1.8, 95% CI 1.6 to 1.9 and cOR 4.14, 95% CI 3.16 to 5.41, respectively). A Canadian study[20] found that the BBA rate trebled over a 16-year period (cOR 3.47, 95% CI 2.31 to 5.20). One UK study[16] found no statistically significant change (cOR 1.28, 95% CI 0.63 to 2.60) and in one French study,[5] there was weak evidence of a small reduction in the adjusted risk of BBA in communities affected by OU closure (adjOR 0.91, 95% CI 0.84 to 1.00).

## Neonatal unit (NNU) admission

One Canadian study[20] suggested a significant reduction in NNU admission following OU closure (cOR 0.39, 95% CI 0.25 to 0.60).

## Apgar score

One Canadian study[20] found no statistically difference in 5 min Apgar score of less than 7 before and after OU closure (cOR 1.13, 95% CI 0.82 to 1.55).

### Hypoxic-ischaemic encephalopathy (HIE)

No studies reported this outcome.

## Evidence from travel distance studies
### Description of included studies

Seven studies described the effect of travel distance to the nearest OU on maternal and neonatal outcomes (table 4). All were published in full between 1991 and 2015. The earliest study[23] was conducted in the UK, three more recent studies were conducted in France,[5 24 25] and one each in Norway,[26] Finland[27] and Canada.[28] Four were retrospective population-based cohort studies, and three were case–control studies. The eligibility criteria varied across studies. Pasquier *et al*[24] included a group with special needs in the form of babies with congenital malformations. Only singleton live births were included in two studies.[5 28]

Travel distance was estimated using geographical mapping software in all studies. However, only three studies[5 24 27] measured the actual distance from women's homes to the nearest OU. In two studies[25 28] a central geographical point for the postal code or municipality was used to estimate distances and in one study the distance was self-reported.[26] Additionally, the studies differed regarding their distance categories, which ranged from 2 to 150 km (table 4).

### Risk of bias assessment

The main risk of bias concerns related to the measurement of exposure, as three studies[25 26 28] did not calculate the distance from the woman's home but used a central point instead or self-reported distance (table 5). Another study[23] measured distance between women's homes and hospital using a straight line. Further risk of bias related to a lack of comparability between study groups in the three case–control studies,[23 26 27] a lack of adjustment for confounders in two studies and missing data in two studies.[26 27] There were no risk of bias concerns relating to sample selection in the cohort studies or outcome measurement.

## Findings
### Maternal outcomes
#### *Maternal mortality (MM)*

In one case–control study from Finland,[27] no maternal deaths were reported in either group (table 6).

#### *Caesarean section (CS) (overall or intrapartum)*

One study from Canada[28] found no statistically significant differences in CS rates with increasing distance (<50 k, 50–150 k, >150 k) based on both crude and adjusted results.

#### *Emergency CS*

The same study from Canada[28] reported no significant difference in emergency CS rates between women living at different distances from an OU based on cORs.

#### *Severe perineal trauma*

No studies reported this outcome.

#### *Postpartum haemorrhage (PPH)*

No studies reported this outcome.

#### *Maternal admission to ICU*

No studies reported this outcome.

#### *Maternal blood transfusion*

No studies reported this outcome.

### Neonatal outcomes
#### Stillbirth (SB) (overall or intrapartum)

Two cohort studies[25 28] reported this outcome. A Canadian study[28] included births to women aged over 35 years who lived <50 km, 50–150 km and >150 km from the OU. SB rates did not change by distance category in the adjusted analysis. A French study[25] reported SB rates at different distances (<5 km, 5–15, 15–30, 30–44, 45+ km) from mother's municipality of residence to the closest OU. The cORs showed women living at 30–44, 15–29 and 5–14 km from an OU had a statistically significantly lower rate of SB compared with women living <5 km from an OU (5–14 km vs <5 km, cOR 0.87, 95% CI 0.85 to 0.90; 15–29 km vs <5 km; cOR 0.85, 95% CI 0.82 to 0.88; 30–44 km vs <5 km, cOR 0.85, 95% CI 0.81 to 0.90). The findings still hold for the adjusted analysis (limited data reported). However, neither the crude nor the adjusted analysis showed a significant difference in risk of SB for individuals resident 45+ km from an OU compared with <5 km.

**Table 4** Description of included studies—travel distance

| Author, year, country | Study design and setting | Study objectives | Study period | Eligibility criteria | Participant characteristics | Description of exposures travel distance | Services context information | Review outcomes — Perinatal | Maternal |
|---|---|---|---|---|---|---|---|---|---|
| **UK Studies** | | | | | | | | | |
| Bhoopalam et al[23], 1991, UK | Case-control study, 2 OUs | To establish BBA prevalence and women at risk of BBA, and morbidity and mortality associated with BBA births | 1983–1987 | Included cases: Women and their BBA babies Included controls: Two controls for each BBA case, one random (next born in the same hospital), one matched (next born matched by GA and BW) Excluded: BW <500g | N(BBAs)=137, 1 twins All n=398 Age yrs (n): <21 (69) 21–35 (339) >35 (27) Nullips (107) Ethnicity (n): European(191) Asian (101) Other (16) SES and education: NR | Distance (km): <2 2–7 >7 | Universal state provision, 2 units six miles apart, serving rural areas of Warwickshire | BBA | NR |
| **Other European Studies** | | | | | | | | | |
| Pasquier et al[24], 2007, France | Retrospective, population-based cohort, 3 Level-III maternity wards with neonatal surgical centre, Rhône-Alpes Region | To examine maternal origin, distance to the nearest maternity ward with a neonatal surgical centre, on perinatal diagnosis, elective termination of pregnancy, delivery in an adequate place and neonatal mortality (NM) for pregnancies with severe malformations | 1990–1995 and 1996–2000 (two periods separated due to changes in prenatal screening) | Included: Fetuses with omphalocele, gastroschisis, diaphragmatic hernia or spina bifida that required surgical repair Excluded: Chromosomal anomalies fetuses and babies without anomalies | n=706 infants n=554 (analysed) Age: yrs (n, %) <21 (15, 2) 21–35 (550, 82)>35 (106, 16) Ethnicity: (n, %) Western European (393, 76) Non-Western European (124, 24) Parity, SES and education: NR | Distance (km): <11 11–50 >50 | Distance to Level III maternity ward with NNU and a neonatal surgical centre, there were three in the Rhone-Alps Region | NM | NR |
| Blondel et al[5], 2011, France* | Retrospective cohort, population based study, metropolitan France | To calculate the incidence of BBA birth in relation to distance from maternity units and the impact of recent closure on different sociodemographic groups | 2005–2006 | Included: Singleton live births Excluded: Municipalities with >8% missing data, unrealistically high BBA births, Departments were excl. if >20% births already excl. | n=1 517 599 livebirths n=1 349 751(analysed) Age yrs - n<20–26 152 20–34 – 105 790 135–39 - 213 534 40+ - 52 164 Parity: Nullip n –774 460 SES n: Occupation professional- 217 045 Intellectual- 325 746 Admin- 266 000 Retail-122,727 Skilled- 149 201 Unskilled- 84 664 None- 84 368 Ethnicity nd education: NR | Distance (km): <5 5–14 15–29 30–44 45+ | Centralising births in larger units | BBA; BBA by parity | NR |

Continued

**Table 4** Continued

| Author, year, country | Study design and setting | Study objectives | Study period | Eligibility criteria | Participant characteristics | Description of exposures travel distance | Services context information | Review outcomes Perinatal | Maternal |
|---|---|---|---|---|---|---|---|---|---|
| Pilkington et al[25], 2014, France | Retrospective, population-based cohort study, French National Vital Statistics registry from mainland France | To investigate the impact of distance to closest maternity unit on perinatal outcomes | 2001–2008, Stillbirth (SB) data 2002–2005 | Included: All births Excluded: NR | n=3 086 128 all births n=3 085 839 (analysed) Age yrs <25–494689 25–34 – 2 008 320 35–39 – 469,975 40+ - 113 144 Singleton pregnancy n=2 988 169 Multiple pregnancy: n=97 959 Parity, ethnicity and socioeconomic status and education: NR | Distance (km): <5 5–14 15–29 30–44 ≥45 | 1998 to 2003, 20% of maternity units closed Mean distance to nearest maternity unit increased (6.6–7.2km) | SB; NM | NR |
| Ovaskainen et al[27], 2015, Finland | Case–control study, one centre, Tampere University Hospital | To establish if BBA births increased over time, to identify risk factors associated with BBAs, also if BBAs babies were more prone to neonatal morbidities compared with those delivered in hospital | 1996–2011 | Included cases: Planned and unplanned BBA Included Controls: 2 controls for infant and mother for each BBA case Excluded: BBA with no information whether planned or unplanned | Cases: BBAs (n=67 births): Age yrs (mean, SD) (range)- 29.0, 5.9 (15–47) Parity 1 (0–16) Controls: n=134 Plurality, ethnicity, education and SES: NR. | Distance (km): <35 ≥35 | Tampere University Hospital is the catchment area for 23 municipalities, 521 700 residents 5000 births/yr | BBA | NR |
| Fougner et al[26], 2000, Norway | Case–control study, 14 municipalities, Oppland County | To compare the experience and care of women who delivered during transport to hospital and women who delivered an hour after arriving to hospital | 1989–1997 | Included cases: Women who delivered their babies before arriving at hospital Included Controls: Women who delivered their babies with 1 hour after arriving at hospital Excluded: NR | n=202 Cases: n=115BBA women Parity n (%)- Nullips 15 (13%) Controls: n=87 women Parity- Nullips 18 (20%) age, ethnicity, education and SES: NR | Distance (km): <12.88 ≥12.88 | Oppland county: 4 hospitals | BBA | NR |
| **Canadian Studies** | | | | | | | | | |
| Lisonkova et al[28], 2011, Canada | Retrospective population-based cohort study, British Columbia | To examine the association between rural residence and birth outcomes in older mothers | 1999–2003 | Included: Singleton mothers aged 35+ Excluded: Women with missing postcodes, babies with congenital anomaly | n=29698 women age >35 years parity n (%): Nullip 87733 (0%) Low SES (n, %) (4385 14% 22.6 vs 3615, 13.7) Ethnicity n (%) first nation 826 (2.8%) Education: NR | Distance (km): <50 50–150 >150 | 17 small maternity units (250–2500 births/yr) closed between 1999 and 2003 | SB; perinatal mortality (PM); NNU admission ≥1 day | All CS; Emergency CS |

*Blondel et al[5] is also included in the OU closure.
adjOR, adjusted OR; BBA, born before arrival; BW, birth weight; CS, caesarean section; ICU, intensive care unit; NHS, National Health Services; NNU, neonatal unit; NR, not reported; OU, obstetric unit; SES, socioeconomic status.

**Table 5** Risk of bias—travel distance

| Author, year, country | Study sample selection bias additional criteria for case–control studies | Bias in measurement of exposure | Bias in measurement of outcomes | Attrition bias | Analysis method reported and appropriate | Potential confounders adjusted for and listed |
|---|---|---|---|---|---|---|
| **…studies** | | | | | | |
| …palam et al[23], UK | Case definition adequate **YES** From hospital records; Representative-ness of cases **YES** BBA cases from 2 hospitals over 5 years; Appropriate selection of controls **YES** two controls from same hospital; Definition of control appropriate **YES** The outcome (BBA) could not have occurred; Comparability of cases and controls **NO** Significant differences in maternal age, parity, ethnicity and antenatal booking | **HIGH** Distance measured in straight line from home address | **LOW** Objective outcome (BBA) | **LOW** 3/134 (2.2%) BBA cases and 10/274 (3.6%) excluded from distance analysis | **HIGH** Descriptive analysis only | **HIGH** No adjustments |
| **European Studies** | | | | | | |
| …uier et al[24], 2007, …ce | **LOW** Data from France Central-East malformation registry | **LOW** GIS software used to estimate distance between maternal residence and nearest maternity ward with neonatal surgical centre | **LOW** Objective outcomes (NM) | **LOW** 12 births (0.03%) missing survival data | **LOW** Method detailed and appropriate, univariate analysis & multiple logistic regression were reported | **LOW** Adjusted for parity, … and other characteristics |
| …del et al[5], 2011, …ce* | **LOW** Birth certificates | **LOW** GIS software used to estimate distance to hospital from home | **LOW** Objective outcome (BBA)* | **LOW** 11% of births excluded | **LOW** Method detailed and appropriate, multilevel model was reported | **LOW** Adjusted for maternal occupation, parity and other characteristics |
| …gton et al[25], 2014, …ce | **LOW** French National Vital Statistics Registry | **HIGH** Distance calculated from centre of municipality not home address, using road networks provided by the French National Geography Institute | **LOW** Objective outcomes (SB, NM) | **LOW** 10% missing for type of pregnancy and 17% for maternal age | **LOW** Method detailed and appropriate, logistic regression analysis was reported | **LOW** Adjusted for maternal… plurality, unemployment ra… single parent households |

**Table 5** Continued

| Author, year, country | Study sample selection bias **additional criteria for case–control studies** | Bias in measurement of exposure | Bias in measurement of outcomes | Attrition bias | Analysis method reported and appropriate | Potential confounders adj... for and listed |
|---|---|---|---|---|---|---|
| ...ner et al[26], 2000, ...nd | Case definition adequate **NO** Data from a questionnaire<br><br>Representative-ness of cases **YES** Cases from three hospitals in one county over 8 years<br><br>Appropriate selection of controls **Yes** Women who delivered their babies within 1 hour of arriving at hospital<br><br>Definition of control appropriate **Yes** Women with no BBA<br><br>Comparability of cases and controls **Unclear** | **HIGH** Data for distance was self-reported in questionnaire | **LOW** Objective outcome (BBA) | **HIGH** Data from women who responded to questionnaire | **HIGH** Descriptive analysis only | **HIGH** No adjustment |
| ...kainen et al[27], Finland | Case definition adequate **YES** Medical records<br><br>Representative-ness of cases **YES** Cases from one centre, but over 15 years<br><br>Appropriate selection of controls **YES** Births occurring immediately preceding and following case<br><br>Definition of control appropriate **YES** The outcome (BBA) could not have occurred<br><br>Comparability of cases and controls **No** Sig. differences for parity, partnership status, smoking, antenatal visits, labour duration and distance to delivery unit | **LOW** Distance from women's home calculated using web-based route planner | **LOW** Objective outcome (BBA) | **High** 13 out-of-hospital deliveries (19%) excluded as could not ascertain whether planned or not planned | **LOW** Method detailed and appropriate, logistic regression data were given | **LOW** Adjusted for single-t... parity, and other characte... |
| **dian studies** | | | | | | |
| ...kova et al[28], Canada | **LOW** Population-based study | **HIGH** GIS used to calculate distance from postcode central point of residence to hospital; mostly using straight line distance | **LOW** Objective outcomes (SB; PM); NNU admission) | **LOW** 492 (1.7%) women excluded due to missing postcodes | **LOW** Method detailed and appropriate, multivariate regression analysis reported | **LOW** Adjusted for parity, ... mother, low income, ethni... other characteristics |

...del et al[5] 2011 included in travel distance & OU closure.
...born before arrival; GIS, Geographical Information System; NM, neonatal mortality; NNU, neonatal unit; PM, perinatal mortality; SB, still birth; sigs, significant.

**Table 6** Outcomes—travel distance

| Outcomes | Author, year, country | Exposure groups | Participants (N, n, %) | | | Findings | | | |
|---|---|---|---|---|---|---|---|---|---|
| **Maternal outcomes** | | | | | | | | | |
| **Maternal mortality** | Ovaskainen et al[27], 2015, Finland | **Travel distance (km):** <35 ≥35 | Groups: <35, ≥35 | N (201): NR, NR | NM n (%): 0, 0 | No events in either group | | | |
| **Caesarean section (CS) (overall or intrapartum)** | Lisonkova et al[28], 2011, Canada | **Travel distance (km):** <50 50–150 >150 | Groups: <50, 50–150, >50 | N (29 698): 27836, 1534, 328 | CS n (%): 9099 (32.70), 464 (30.25), 94 (28.70) | Groups: <50, 50–150, >50 | | Crude OR (95% CI): 1, 0.89 (0.80 to 1.00), 0.83 (0.65 to 1.05) | Adjusted OR (95% CI): NR, NR, NR |
| **Emergency CS** | Lisonkova et al[28], 2011, Canada | **Travel distance (km):** <50 50–150 >150 | Groups: <50, 50–150, >50 | N (9657): 9099, 464, 94 | Emergency CS n (%): 5378 (59.11), 258 (55.60), 52 (55.32) | Groups: <50, 50–150, >50 | | Crude OR (95% CI): 1, 0.87 (0.72 to 1.05), 0.86 (0.57 to 1.29) | Adjusted OR (95% CI): NR, NR, NR |
| **Severe perineal trauma** | No studies | | | | | | | | |
| **Postpartum haemorrhage** | No studies | | | | | | | | |
| **Admission to ICU** | No studies | | | | | | | | |
| **Blood transfusion** | No studies | | | | | | | | |
| **Neonatal outcomes** | | | | | | | | | |
| **Stillbirth (SB)** | Pilkington et al[25], 2014, France | **Travel distance (km):** <5 5–14 15–29 30–44 ≥45 | Groups: <5, 5–14, 15–29, 30–44, ≥45 | N (30 859) (2002–2005): 1 404 665, 811 775, 648 495, 186 537, 34 367 | SB n (/per 1000): 13204 (9.4), 6657 (8.2), 5188 (8.0), 1492 (8.0), 306 (8.9) | Groups: <5, 5–14, 15–29, 30–44, ≥45 | | Crude OR (95% CI): 1, 0.87 (0.85 to 0.90), 0.85 (0.82 to 0.88), 0.85 (0.81 to 0.90), 0.95 (0.84 to 1.06) | Adjusted OR (95% CI): 1, Reported as RR 0.87 (NR)*, Reported as RR 0.85 (NR)*, Reported as RR 0.85 (NR)*, Reported as RR 0.95 (NR)(NS) |
| | Lisonkova et al[28], 2011, Canada | **Travel distance (km):** <50 50–150 >150 | Groups: <50, 50–150, >150 | N (29 698): 27836, 1534, 328 | NM n (%): 150, NR, NR | Groups: <50, 50–150, >150 | | Crude OR (95% CI): NR, NR, NR | Adjusted OR (95% CI): NR, NR, NR |

OR NR. Authors noted SB rate was higher among women living 50–150km and >150km vs <50km, no significant difference found after adjusting for confounders.

Continued

**Table 6** Continued

| Outcomes | Author, year, country | Exposure groups | Participants (N, n, %) | | Findings | | | |
|---|---|---|---|---|---|---|---|---|
| | | | | | Groups | NM n (%) | Crude OR (95% CI) | Adjusted OR(95% CI) |
| **Neonatal mortality (NM)** | Pasquier et al[24], 2007, France | **Travel distance (km): <11** 11–50 >50 | **N (554)** | | <11 | NR | NR | 1 |
| | | | 239 | | 11–50 | NR | NR | 0.98 (0.34 to 2.88) |
| | | | 156 | | >50 | NR | NR | 1.37 (0.49 to 3.86) |
| | | | 159 | | | | | |
| | Pilkington et al[25], 2014, France | **Travel distance (km): <5** 5–14 15–29 30–44 ≥45 | **N (6 202 918) (2001–2008)** | | Groups | NM n (/per 1000) | Crude OR (95% CI) | Adjusted RR (95% CI) |
| | | | 2 808 068 | | <5 | 7582 (2.7) | 1 | 1 |
| | | | 1 626 885 | | 5–14 | 3416 (2.1) | 0.78 (0.75 to 0.81) | Reported as RR 0.91 (NR) * |
| | | | 1 316 329 | | 15–29 | 2896 (2.2) | 0.81 (0.78 to 0.85) | Reported as RR 0.94 (NR)(NS) |
| | | | 381 288 | | 30–44 | 801 (2.1) | 0.78 (0.72 to 0.84) | Reported as RR 0.9 (NR)* (NR)(NS) |
| | | | 69 787 | | ≥45 | 154 (2.2) | 0.82 (0.70 to 0.96) | Reported as RR 0.96 (NR)(NS) |
| | | **NM after BBA** | **NM after BBA** | | **NM after BBA** | | | |
| | | Groups | Groups | | Groups | NM n (/per 100,000) | Crude OR (95% CI) | Adjusted RR (95% CI) |
| | | <5 | 2 808 068 | | <5 | 115 (4.1) | 1 | 1 |
| | | 5–14 | 1 626 885 | | 5–14 | 65 (4.0) | 0.98 (0.72 to 1.32) | Reported as RR 1.1 (NR) (NS) |
| | | 15–29 | 1 316 329 | | 15–29 | 72 (5.5) | 1.34 (0.99 to 1.79) | Reported as RR 1.58 (NR)* |
| | | 30–44 | 381 288 | | 30–44 | 23 (6.0) | 1.47 (0.94 to 2.30) | Reported as RR 1.51 (NR)(NS) |
| | | ≥45 | 69 787 | | ≥45 | 7 (10.0) | 2.45 (1.14 to 5.25) | Reported as RR 3.68 (NR)* |
| **Perinatal mortality (PM)** | Lisonkova et al[28], 2011, Canada | **Travel distance (km): <50** 50–150 >150 | **N (29 698)** | | Groups | PM n (%) | Crude OR (95% CI) | Adjusted OR (95% CI) |
| | | | 27 836 | | <50 | 221 (0.80) | 1 | 1 |
| | | | 1534 | | 50–150 | 19 (1.24) | 1.57 (0.98 to 2.51) | 1.53 (1.10 to 2.12) |
| | | | 328 | | >50 | 8 (2.44) | 3.12 (1.53 to 6.38) | 3.06 (2.20 to 4.24) |
| **Infant mortality (IM)** | No studies | | | | | | | |

Continued

**Table 6** Continued

| Outcomes | Author, year, country | Exposure groups | Participants (N, n, %) | Groups | Findings | | |
|---|---|---|---|---|---|---|---|
| **Born before arrival (BBA)** | Bhoopalam et al[23], 1991, UK | **Travel distance (km):** <2 2–7 >7 | N (398) cases and controls | Groups | BBA cases n (%) | Crude OR (95% CI) | Adjusted OR (95% CI) |
| | | | 59 | <2 | 4 (6.80) | 1 | NR |
| | | | 249 | 2–7 | 88 (35.34) | 7.52 (2.64 to 21.43) | NR |
| | | | 90 | >7 | 42 (46.70) | 12.03 (4.02 to 36.01) | NR |
| | Blondel et al[5] 2011, France | **Travel distance (km):** <5 5–14 15–29 30–44 >45 | N (1 359 756) | Groups | BBA n (rate /1000 births) | Crude OR (95% CI) | Adjusted OR (95% CI) |
| | | | 596363 | <5 | 1849 (3.1) | 1 | NR |
| | | | 352279 | 5–14 | 1395 (3.9) | 1.28 (1.19 to 1.37) | NR |
| | | | 296734 | 15–29 | 1659 (5.6) | 1.81 (1.69 to 1.93) | NR |
| | | | 88670 | 30–44 | 692 (7.8) | 2.53 (2.32 to 2.76) | NR |
| | | | 15705 | >45 | 182 (11.) | 3.77 (3.23 to 4.39) | NR |

| | | Groups | Parity 1 nd 2 n=152 426 Adjusted OR (95% CI) | Parity 3+ N=197325 Adjusted OR (95% CI) |
|---|---|---|---|---|
| | | <5 | 1 | 1.73 (1.57 to 1.90)a |
| | | 5–14 | 1.14 (1.03 to 1.27) | 2.32 (2.04 to 2.63) |
| | | 15–29 | 1.39 (1.24 to 1.57) | 3.25 (2.84 to 3.71) |
| | | 30–44 | 1.78 (1.55 to 2.05) | 3.71 (3.13 to 4.41) |
| | | >45 | 2.47 (2.02 to 3.02) | 6.46 (4.92 to 8.48) |

| Outcomes | Author, year, country | Exposure groups | Participants | Groups | BBA | Crude OR (95% CI) | Adjusted OR (95% CI) |
|---|---|---|---|---|---|---|---|
| | Ovaskainen et al[27], 2015, Finland | **Travel distance (km):** <35 ≥35 | N (201) | Groups | BBA n (%) | Crude OR (95% CI) | Adjusted OR (95% CI) |
| | | | 67 | BBA cases | | | |
| | | | 134 | Controls | | | |
| | | | NR | <35 | NR | NR | 1 |
| | | | NR | ≥35 | NR | NR | 5.02 (1.80 to 14.04) |
| | Fougner et al[26], 2000, Norway | **Travel distance (km):** <12.88 ≥12.88 | N (202) cases and controls | Groups | BBA n (%) | Crude OR (95% CI) | Adjusted OR (95% CI) |
| | | | 90 | <12.88 | 44 (48.90) | 1 | NR |
| | | | 112 | ≥12.88 | 71 (63.34) | 1.81 (1.03 to 3.18) | NR |
| **Neonatal unit admission (NNU)** | Lisonkova et al[28], 2011, Canada | **Travel distance (km):** <50 50–150 >150 | N (15 325) | Groups | NNU n (%) | Crude OR (95% CI) | Adjusted OR (95% CI) |
| | | | 14333 | <50 | 648 (4.80) | 1 | NR |
| | | | 815 | 50–150 | 32 (3.92) | 0.86 (0.60 to 1.24) | NR |
| | | | 177 | >150 | 12 (6.80) | 1.54 (0.85 to 2.77) | NR |
| **Apgar score** | No studies | | | | | | |
| **HIE** | No studies | | | | | | |

aSignificant difference.
†Reference group: women with 1&2 parity and <5 km.
CS, caesarean section; GA, gestational age; HIE, hypoxic-ischemic encephalopathy; ICU, intensive care unit; NNU, neonatal unit; NR, not reported; NS, not significant; OR, odd ratio; RR, relative risk.

### Neonatal mortality (NM)

Two French cohort studies[24 25] reported this outcome. One study[24] examined the distance from women's homes to the nearest OU with neonatal surgical facilities for 706 fetuses with severe malformations. Analyses adjusted for malformation type, number of malformations, amniotic fluid anomaly, previous anomaly in the family and parity showed no association between NM and distance (<11 km vs 11–50 km, adjOR=0.89, 95% CI: 0.34, 2.88; <11 km vs >50 km, adjOR=1.37, 95% CI: 0.49, 3.86). The other study[25] included all births and found that NM rates were significantly higher for women living <5 km compared with 5–44 km away from an OU[25] (5–14 km vs <5 km, cOR 0.78, 95% CI: 0.75, 0.81; 15–29 km vs <5 km cOR 0.81, 95% CI: 0.78, 0.85; 30–44 km vs <5 km, cOR 0.78, 95% CI: 0.72, 0.84; ≥45 km vs <5 km, cOR 0.82, 95% CI: 0.70, 0.96). In this latter study, the NM of babies BBA was also explored. For the BBA group, there was a statistically significant increase in the risk of NM when women had to travel 45 km or more to an OU in comparison to <5 km (≥45 km vs <5 km, cOR 2.45, 95% CI 1.14 to 5.25).

### Perinatal mortality (NM)

A study from Canada[28] reported that PM risk increased with travel distance in an adjusted model (<50 km v 50–150 km adjOR 1.53, 95% CI 1.1 to 2.12; <50 km >150 km adjOR 3.06, 95% CI 2.20 to 4.24).

### Infant mortality (IM)

No studies reported this outcome.

### Born before arrival (BBA)

Three case–control studies,[23 26 27] and one cohort study[5] reported this outcome. All four studies reported a significant increase in BBA rate with longer travel distance, although only two reported adjusted analyses.[5 27] In the UK study,[23] the risk of BBA increased 12-fold for women living >7 km from the OU compared with women living <2 km away (cOR 12.5, 95% CI 4.02 to 36.01). The risk of BBA increased significantly for women living >13 km from an OU in a Norwegian study[26] (cOR 1.81, 95% CI 1.03 to 3.18). The Finnish study[27] reported a fivefold increased risk of BBA for women living >35 km from the OU compared with <35 km (adjOR 5.02, 95% CI 1.80 to 14.04).

In France,[5] the rate of BBA significantly increased with longer distances and it tripled for all women living 45+ km from the OU compared with women living <5 km away (cOR 3.77, 95% CI 3.23 to 4.39). The association persisted in an adjusted analysis which included women of parity three or higher and living 45+ km from the OU, who had a sixfold increased risk of BBA compared with women living <5 km away and of parity one or two (adjOR 6.49, 95% CI 4.92 to 8.48).

### Neonatal unit (NNU) admission

A study from Canada[28] reported an increase in NNU admission for births to women living >150 km from an OU compared with those living <50 km away (6.8% vs 4.8%).

### Apgar score

No studies reported this outcome.

### Hypoxic-ischaemic encephalopathy (HIE)

No studies reported this outcome.

### Evidence from travel time studies
#### Description of included studies

Fifteen studies explored the impact of travel time from a woman's home to an OU (see table 7). Two studies (one reported as an abstract only) were conducted in the UK,[29–31] three studies in France,[32–34] three studies (reported in five articles) in the Netherlands,[35–39] one study reported in two articles from Norway,[40 41] five studies in Canada[42–46] and one study in Japan.[47]

Eleven studies were of a retrospective cohort design, one was a prospective cohort study,[39] one was a before-and-after design[47] and two were case–control studies.[33 34] All the studies clearly stated the eligibility criteria. Only singleton births were included in five studies.[30–32 35–38 42] One study[39] specifically enrolled women with postnatal haemorrhage after home birth, and one study[42] focused on planned home birth regardless of the actual place of birth.

The studies were heterogeneous in their travel time intervals. With the exception of one study in Canada,[42] longer time cut-off points were examined in studies from Norway, Japan and Canada compared with studies in other countries (all European). Travel duration was estimated using geographical mapping software in all studies. However, most studies estimated travel duration to and from central points within areas rather than actual addresses.

#### Risk of bias assessment

Risk of bias assessment and supported explanations for each of the risk of bias domains are presented in table 8. With the exception of Stolp et al,[39] sample selection and measurement of outcomes were considered to be at low risk of bias across all studies as such data were obtained from national databases and birth registries. The groups in the two case–control studies were appropriately selected and defined, however, the case and control groups were not comparable in both studies (eg, difference in antenatal care attendance and sociodemographics). Eight studies[29 34–42] were considered at low risk of exposure measurement bias, as the women's actual place of residence was used to estimate travel time to nearest OU. The risk of attrition bias was low for the majority of the included studies. Similarly, analyses and adjustment for potential confounders were found to be appropriate in the majority of studies.

### Findings
### Maternal outcomes

### Maternal mortality (MM):
No studies reported this outcome.

**Table 7** Description of included studies— travel time

| Author, year, country | Study design and setting | Study objectives | Study period | Eligibility criteria | Participant characteristics | Description of exposures travel distance | Services context information | Review outcomes Perinatal | Maternal |
|---|---|---|---|---|---|---|---|---|---|
| **UK Studies** | | | | | | | | | |
| Dummer et al[29], 2004, UK | Retrospective population-based cohort study, Cumbria | To investigate whether geographical accessibility to hospitals affected SB rates and infant mortality | 1950–1993 grouped: 1950–1959 1960–1969 1970–1979 1980–1993 | Included: All births Excluded: Women with missing postcodes | n=283 668 births Other characteristics: NR | Travel time (mins): <17 18–35 >35 | Universal state provision of maternity care, 1950–1993: 4 hospitals opened, 2 closed | NM; Early NM; Post NM | NR |
| Paranjothy et al[30][31], 2013, 2014, UK (abstract& full paper) | Retrospective cohort study, All Wales Perinatal Survey and National Community Child Health Database | To study the association between travel time from home to OU on intrapartum stillbirth (SBJandNM) | 1995–2009 | Included: All registered birth >23 wks GA Excluded: Antepartum SB, lethal congenital anomalies, multiple pregnancies, invalid or missing GA, missing maternal age/postcode/ hospital of birth or baby's gender | n=466 255 singleton births Maternal age yrs %<20 90.7 20–34 76.5 34–44 13.8 45=0.1 Parity: Nullips 44.9% Social deprivation quintile %: 1 (least depr) 16.7 2–4 57.8 5 (most depr) 25.6 Ethnicity, education: NR | Travel time (mins): <15 15–29 30–44 >45 | Universal state provision of maternity care. 50 hospitals (16 outside Wales) | Intrapartum SB; Early NM; Late NM | NR |
| **Other European Studies** | | | | | | | | | |
| Combier et al[32], 2013, France | Retrospective cohort study, based on hospital discharge summaries, Burgundy | To analyse the effect of travel time to closest OU on pregnancy outcome and prenatal management in Burgundy | 2002–2009 | Included: Singleton births >21 wks GA Excluded: Medical ToP, multiple pregnancy, births outside Burgundy, births in 2002 and 2008 due to closure of 3 units | n=111 001 births Other characteristics=NR | Travel time (mins): ≤15 16–30 31–45 ≥46 | 2000–2001: 2 private maternity units closed 2002–2009: 3 public maternity units closed. Units(n); 2000 (20) 2009 (15) | SB; PM; BBA | NR |
| Renesmeet al[34], 2013, France | Case–control, multicentre study, 8 units, Finistere District, Brittany | To evaluate the social-geographical factors associated with BBAs | 2007–2009 | Included cases: BBA of live birth Included controls: 2 controls for each case irrespective of delivery mode. Excluded: GA <22 weeks, BW <500 g, planned home birth | n=225 Cases vs controls n=76 vs 149 Age (median, range) yrs: 30 (16–41) vs 30 (16–41) Parity (median, range): 2 (1–6) Maternal INSEE code n (%): 1, 2, 3 or 4=15 (23.8) vs 56 (43.4); five or 6: 20 (31.8) vs 55 (42.6); 8=28 (44.4%) vs 18(14) Ethnicity, education: NR | Travel time (mins): 15 15–29 30–44 >45 | 9700 births/year in Finistere In 2012 units with <300 births/yr were closed. Universal state provision of maternity care | BBA | NR |
| Nguyen et al[33], 2016, France | Case–control study, university hospital in Caen | To estimate the incidence of BBA during the study period | 2002–2009 | Included cases: Unplanned BBA Included controls: Next spontaneous birth in hospital Excluded: NR | n=188 Cases n=94 Mean age: 28.9 years Parity: 1.8 SES: 73.4% no profession/student Control n=94 Mean age: 29.2 years Parity: 0.9 SES: 47.9% no profession/ student Ethnicity, education: NR | Travel time (mins): ≤20 mins >20 mins | University Hospital with neonatal care facilities. Universal state provision of maternity care | BBA | NR |
| **Ravelli Study** | | | | | | | | | |

Continued

**Table 7** Continued

| Author, year, country | Study design and setting | Study objectives | Study period | Eligibility criteria | Participant characteristics | Description of exposures travel distance | Services context information | Review outcomes Perinatal | Maternal |
|---|---|---|---|---|---|---|---|---|---|
| Ravelli et al[35–37], 2011, Netherlands (full papers & abstract) | Retrospective population-based cohort study, rural and urban areas, 12 provinces | To study the effect of travel time from home to OU on mortality and other adverse outcomes in pregnant women at term in primary and secondary care | 2000–2006 | Included: Singleton term births Excluded: Antepartum deaths, congenital anomalies, invalid/missing postcodes or outpatient codes, or births from Wadden islands, home deliveries, hospitals participated for 1–2 years | n=751 926 singleton births Age yrs, % <20, 2 20–34, 78.3 35–39, 17.2 ≥40. 2.4 Parity: Nullips: 49.9% Ethnicity: White 81.7% SES % : high 25.2 medium 48.2, low 26.7 Education: NR | Travel time (mins): <20≥20 | Universal state provision of maternity care. 99 OUs including tertiary perinatal centres | NM (Combined intrapartum & early & late NM up to 28 days) NM (0–24 hours) NM (0–27 days) NM(8–27 days); Combined (mortality and/or Apgar<4 at 5 min, and/ or NNU admission) | NR |
| Ravelli et al[38], 2012, Netherlands | Retrospective cohort study in nine regions | To investigate provincial differences in perinatal mortality (PM) and to determine the influence of different risk factors, including travel time from home to the OU during labour | 2000–2006 | Included: Singleton births Excluded: Women with incorrect post codes | n=1 242 725 singletons Age yrs, % <20, 1.8>35, 19.5 Parity, % Nullips, 46.3 Ethnicity, % Non-western 16.2 SES low (10th centile): 10% Education NR | Travel time(mins): <20≥20 | Universal state provision of maternity care | PM | NR |
| Stolp et al[39], 2015, Netherlands | Prospective cohort study, rural & urban areas | To assess whether the limit of 45 mins is met for ambulance transfer of women with PPH after home birth | 2008–2010 | Included: Women with PPH after MW supervised home birth Excluded: Cases of PPH with missing data | n=72 (54 analysed) Age median (range) yrs: 31 (23–41); Parity (n, %); Primip 27%–50% Ethnicity, Education, SES: NR | Travel time (mins): <45 >45 | Home birth for low risk women and hospital birth. Universal state provision of maternity care | NR | Maternal admission to intensive care; (ICU); Blood transfusion; Postpartum haemorrhage (PPH) |
| **Egjom Study** | | | | | | | | | |
| Engjom et al[40 41] 2017 & 2015, Norway (full paper & abstract) | Retrospective population-based cohort study, Medical Birth Registry of Norway and Statistics Norway, 19 counties | To assess peripartum mortality associated with place of birth and availability of obstetric units. | 1999–2009 | Included: All births in Norway with GA ≥22 wks or BW ≥500 g Excluded: Lack of address and municipality, antepartum SB, planned home births | n=646 898 960.4% singletons. Age yrs, % <20 20.4 20–35, 80.7>35, 16.9%: Multips 58.7%; Education >11 y 77.2 Ethnicity: Western 90.7% | Travel time (hrs):<1 1–2 >2 | Basic obstetric care for normal delivery; Emergency obstetric care <1500 births/ yr. Universal state provision of maternity care | BBA | NR |
| **Canadian Studies** | | | | | | | | | |
| **Grzybowskind Stroll Study** | | | | | | | | | |
| Grzybowski et al[43], 2011, Canada | Retrospective cohort study, rural areas of British Columbia | To document newborn and maternal outcomes in relation to travel time to the nearest OU with CS capability | 2000–2004 | Included: All deliveries>20 weeks' GA Excluded: Multiple birth, congenital anomalies or late ToP, core urban areas | n=35 426 birthsGroups:<1 hour, 1–2, 2–4,>4 hours Group N: 32 814, 1359, 747, 506 Mean maternal age yrs: 28.7, 28.67, 27.25, 27.2 Parity % primips: 42.6, 38.6, 36.7, 36.8 SES*: 0.12, 0.10, 0.30, 0.33, first Nations % 0.05, 0.30, 0.23, 0.42 Education: NR | Travel time (hrs): <1 1–2 2–4 >4 | Universal medical coverage for core healthcare, 13 NNUs, 42 000 births /year | PM (SB & early NND); BBA; NNU admission | CS |
| Grzybowskiet al[44], 2013, Canada | Retrospective cohort study, rural areas of British Columbia | To compare rural maternity care by level of services | 2000–2007 | Included: Singleton births Excluded: Women with residential postcode of large urban centres | n=4672 births; Mean age, yrs: 27.7 Parity: primips: 39.7% SES*: 0.22%; Ethnicity: first Nations 0.3% Education: NR | Travel time (hrs):<1 >1 | Universal medical coverage for core healthcare | SB; NND (late<1 month); PM; IM; BBA; NNU admission | CS; Emergency CS; PPH |

## Table 7 Continued

| Author, year, country | Study design and setting | Study objectives | Study period | Eligibility criteria | Participant characteristics | Description of exposures travel distance | Services context information | Review outcomes Perinatal | Maternal |
|---|---|---|---|---|---|---|---|---|---|
| Grzybowski et al[45], 2015, Canada | Retrospective cohort study, British Columbia (BC), Alberta, Nova Scotia (NS) | To examine the safety of rural Canadian maternity services | 2003–2008 | Included: Singleton deliveries Excluded: Multiple births, infants born with congenital anomalies, planned home births, accidental BBA | Alberta, BC, NS Age yrs (n %) <18: 1618 (2.3), 1256 (2.0), 413 (2.2) >35 yrs: 5127 (7.3), 8866 (14.3), 2387 (12.7) Multips n (%) 41 730 (59.6), 35 089 (56.6), 10 656 (56.8) Ethnicity, SES, education: NR | Travel time (hrs);<1 1–2 2–4 >4 | Universal medical coverage for core healthcare, 20 small maternity closures since 2000 | PM (SB & NND up to 7 days) | CS |
| Stol et al[46], 2014, Canada | Retrospective cohort study, rural British Columbia | To report on characteristics and perinatal outcomes of rural women with only MW involved in care | 2003–2008 | Included: Women residing outside core urban areas, singletons >20wk GA and care by a MW Excluded: Late ToP, congenital anomalies | <1 hour, 1–2 hours, >2 hours:n=3438, 124, 130 Mean age yrs: 29.78, 31.4, 30.5 Primips n (%) 1574 (45), 63 (50.8), 63 (48.5) Ethnicity, SES & Education: NR | Travel time (hrs); <1 1–2 >2 | Universal medical coverage for core healthcare, closure of 22 rural maternity services | PM (SB & NND up to 7 days) | CS |
| Darling et al[42], 2019, Canada | Retrospective population-based cohort study, Ontario | Whether greater diving distances to OU associated with a higher risk of adverse neonatal outcomes | 2012–2015 | Included: Women who planned home births regardless of actual place of births Excluded: Multiple births, Preterm <37 wks | n=11 869 Age yrs, %: <25, 9.5 25–39,87.6≥40, 2.9 Primps n (%) 4208 (35.5) SES low, n(%) 2465 (20.8) Ethnicity and education: NR | Travel time (mins); ≤30 >30 | Universal medical coverage for core healthcare | PM (PM); NNU admission; 5mins Apgar<7 | CS |
| **Other countries** | | | | | | | | | |
| Aoshima et al[47], 2011, Japan | Before and after study design, data from perinatal care centres | Whether reducing travel time influences the neonatal mortality rate (NM) | 2002–2006 | Included: All births Excluded: Municipalities consisting of isolated islands | Number of births: 2002=347 284 2006=322 514 Other characteristics: NR | Travel time (hrs); ≤1 >1 | Universal healthcare insurance system, 346 perinatal care centres | NM | NR |

INSEE: Institute National de la Statistique et des Etudes Economiques; INSEE codes: 1: farmer; 2: craftsperson, merchant or entrepreneur; 3: businessexecutive, intellectual occupation; 4: other professionals; 5: employee; 6: worker; 8: no occupation.
*SES: Catchment level Social vulnerability −1 to +1
†GPESS = general practitioner with enhanced surgical skills.
BBA, born before arrival; BW, birth weight; CS, caesarean section; GA, gestational age; ICU, intensive care unit; NNU, neonatal unit; NR, not reported; NS, not significant; OU, obstetric unit; RR, relative risk; SES, socioeconomic status; ToP, termination of pregnancy.

**Table 8** Risk of bias—travel time

| Author, year, country | Study sample selection bias / additional criteria for case-control studies | Bias in measurement of exposure | Bias in measurement of outcomes | Attrition bias | Analysis method reported and appropriate | Potential confounders adjusted for and listed |
|---|---|---|---|---|---|---|
| **UK Studies** | | | | | | |
| Dummer & Parker[29], 2004, UK | LOW Cumbrian Births Database | LOW Modelled using GIS | LOW Objective outcome (NM) | LOW Of 3352 live births, 42 stillbirths excluded as the outcome NM | LOW Method detailed, results of LR were reported | LOW Adjusted for year of birth, social class, birth order, multiple births |
| Paranjothy et al[30 31], 2013 & 2014, UK | LOW National Community Child Health Database & All Wales Perinatal Survey) | HIGH Women's address replaced by population-weighted centroid, travel time calculated using Google Maps API (v3) | LOW Objective outcomes (SB, NM) | LOW 11% excluded where information on parity was missing | LOW Analysis method described and multilevel LR data were reported | LOW Adjusted for maternal age, parity, urban/rural location, SES, and other characteristics |
| **European Studies** | | | | | | |
| Combier et al[32], 2013, France | LOW Burgundy perinatal network database | HIGH Municipality town hall not woman's home address | LOW Objective outcomes (SB, PM, BBA) | LOW All births identified included in the analysis | LOW Method described; hierarchical LR and multilevel LR reported | LOW Adjusted for maternal age, urbanisation level and other characteristics |
| Renesme et al[34], 2013, France | Case definition — YES linked to perinatal network database; Representativeness of cases — YES All cases in defined period; Appropriate selection of controls — YES Controls chosen randomly from same databases and from births occurring at the nearest delivery date and hour to cases; Definition of control appropriate — YES Outcome could not have occurred; Comparability of cases and controls — NO Difference in antenatal care attendance | LOW Distance & travel time estimated using GIS | LOW Objective outcomes retrieved from regional and hospital databases | LOW 5/81 (6%) BBAs missing, 3/162 (2%) controls missing | LOW Method described and appropriate; multivariate reported | LOW Adjusted for age, family status, INSEE maternal occupation, parity, and other characteristics |
| Nguyen et al[33], 2016, France | Case definition — YES Using medical records; Representativeness of cases — YES All cases in defined period; Appropriate selection of controls — YES Next birth, of equivalent GA; Definition of control appropriate — YES Outcome could not have occurred; Comparability of cases and controls — NO Significant differences in parity, smoking, pregnancy monitoring, profession | UNCLEAR No information | LOW Objective outcome (BBA) | UNCLEAR No information | HIGH No details of the analysis method and analysis was only descriptive | HIGH No adjustment for any potential confounders |
| Ravelli et al[35–37], 2011, Netherlands (abstract & full papers) | LOW Population based study using the Netherlands Perinatal Registry | LOW GIS software used to measure travel time from women's postcodes | LOW Objective outcomes from perinatal registry | LOW Small proportion (0.3%) of women excluded due to incorrect zip code | LOW Method reported; descriptive analysis & LR results given | LOW Analysis adjusted for age, parity, ethnicity, SES |
| Ravelli et al[38], 2012, Netherlands | LOW Population based study using the Netherlands Perinatal Registry | LOW GIS software used to measure travel time from women's postcodes | LOW Objective outcomes from perinatal registry | LOW Small proportion 4% of women excluded | UNCLEAR No information | LOW Adjusted for age, parity, very urban /very rural, SES |
| Stolp et al[39], 2015, Netherlands | HIGH Study participants were selected by midwives | LOW Ambulance interval includes total time from dispatch call to arrival at hospital | UNCLEAR Method of measuring blood loss not reported | HIGH Missing data 18/72 (25%) due to incomplete documentation | HIGH Data only analysed descriptively | HIGH No adjusted analysis |
| Engjom et al[40], 2017 and Engjom et al[41], 2015, Norway (abstract & full paper) | LOW Medical Birth Registry of Norway | LOW Travel time polygon from home address using GIS | LOW Objective outcomes from birth registry | UNCLEAR No information | LOW Analysis appropriate, details of LR, multilevel modelling were reported | LOW Adjusted for maternal age, parity, education, ethnicity |

Continued

**Table 8**  Continued

| Author, year, country | Study sample selection bias **additional criteria for case–control studies** | Bias in measurement of exposure | Bias in measurement of outcomes | Attrition bias | Analysis method reported and appropriate | Potential confounders adjusted for and listed |
|---|---|---|---|---|---|---|
| Grzybowski et al[43], 2011, Canada | **LOW** Population based study using British Columbia Perinatal Health Programme | **HIGH** GIS used to create 1 hour travel zone for each maternity service, but central postal code to the nearest maternity care used | **LOW** Objective outcomes from Perinatal Health Programme | **LOW** 0.3% excluded due to incorrect zip code | **LOW** Analysis appropriate, descriptive analysis & hierarchical LR reported | **LOW** Adjusted for maternal age, parity, SES, ethnicity |
| Grzybowski et al[44], 2013, Canada | **LOW** Data from Perinatal Data Registry | **HIGH** Community central postal code used not women's home address | **LOW** Objective outcomes from Perinatal Data Registry | **HIGH** Number of women excluded due to incorrect postal address not reported | **LOW** Analysis appropriate descriptive analysis & LR | **LOW** Adjusted for maternal age, parity, lone parent status, ethnicity, SES |
| Grzybowski et al[45], 2015, Canada | **LOW** Provincial perinatal registries | **HIGH** Community central point postal code used not women's home address | **LOW** Objective outcomes from Perinatal Data Registries | **UNCLEAR** No information on missing data | **LOW** Analysis appropriate, descriptive analysis & LR reported | **LOW** Adjusted for maternal age, parity |
| Stoll et al[46], 2014, Canada | **LOW** Based on British Columbia Perinatal Database Registry | **LOW** Used GIS and Google maps; travel times were adjusted for travel conditions | **LOW** Objective outcomes (CS) | **LOW** No missing data | **HIGH** Data were only analysed descriptively | **HIGH** No adjusted analysis |
| Darling et al[42], 2019, Canada | **LOW** Data from Perinatal Registries | **LOW** Driving time from women's residence using online mapping tool ArcGIS | **LOW** Objective outcomes from Perinatal Data Registries | **LOW** 3.7% excluded not being able to calculate distance to nearest hospital | **LOW** Method reported, results of descriptive analysis & LR reported | **LOW** Adjusted for maternal age, parity, gestational age, season, SES |
| Aoshima et al[47], 2011, Japan | **LOW** All Japan except for isolated islands outside road network (96.6% of all Medical Service Areas) | **HIGH** Used central point of municipality not home address but analysis based on (larger) Medical Service Areas | **LOW** Objective outcomes from Medical Service Area databases | **UNCLEAR** No information on missing data | **LOW** Method appropriate, unpaired t-test, difference-in-difference analysis | **HIGH** No adjusted analysis |

BBA, born before arrival; CS, caesarean section; GA, gestational age; GIS, geographical information system; INSEE, institute national de la statistique et des etudes economiques; LR, logistic regression; NM, neonatal mortality; NNU, neonatal unit; PM, perinatal mortality; SES, socio economic status; SB, still birth.

## Caesarean section (CS) (overall, or intrapartum)

Five Canadian studies[42–46] reported CS rates (table 9). Across three studies,[43–45] cORs for CS rates were higher among women who lived closer to OUs with CS rates highest for women living less than 1 hour away compared with other categories (1–2 hours, 2–4 hours and >4 hours). One study[46] included women who had a midwife involved in their care, and found no significant differences in CS rates for women living 1–2 hours and more than 2 hours away compared with within 1 hour of an OU (1–2 vs <1 hour, cOR 1.23, 95% CI 0.80 to 1.91 and >2 hours vs <1 hour, cOR 1.11, 95% CI 0.71 to 1.72). A further study[42] also showed a higher CS rate among women who planned a home birth and lived less than half an hour away from OU services (>30 min vs ≤30 min, cOR 0.74, 95% CI 0.59 to 0.92).

## Emergency CS

Shorter travel time to an OU was associated with a statistically significant higher emergency CS rate in one Canadian study[45] (>1 hour vs <1 hour, cOR 0.80, 95% CI 0.75 to 0.86).

## Severe perineal trauma

No studies reported this outcome.

## Postpartum haemorrhage (PPH)

One Canadian study found the risk of PPH was significantly higher for women who lived more than 1 hour away from obstetric services compared with women who lived less than 1 hour away[44] (>1 hour vs <1 hour, cOR 1.27, 95% CI 1.13 to 1.43).

## Maternal admission to ICU

One study from the Netherlands[39] involved women who had a PPH after midwifery-supervised home births and examined adverse maternal outcomes associated with travel time longer than 45 min to hospital. No difference was found in the number of women admitted to ICU who travelled more than 45 min compared with <45 travel time to hospital, but the numbers of events were low.

## Maternal blood transfusion

One study from the Netherlands[39] found no significant difference in the median number of units of blood transfused to women who travelled more than 45 min to an OU compared with <45 min travel time.

## Neonatal outcomes:

## Stillbirth (SB) (overall or intrapartum)

Three studies examined the association between increasing travel time and SB, one study each from the UK,[30 31] France[32] and Canada.[44]

In the UK study,[30 31] there was no association between travel time and SB when analysing all women (adjOR 1.13, 95% CI 0.98 to 1.30). However, subgroup analyses showed a significant increase in the risk of SB with every 15 min increase in travel time to the OU for term pregnancies (adjOR 1.36, 95% CI 1.17 to 1.59) and for nulliparous women (adjOR 1.21, 95% CI 1.02 to 1.44). The other two studies[32 44] found no significant increase in the incidence of SB with increasing travel time.

## Neonatal mortality (NM)

Five studies examined the association between travel time and NM, two from the UK,[29–31] one from the Netherland,[35–37] one from Canada[44] and one from Japan.[47]

The adjusted analysis in one UK study[29] showed no statistically significant association between NM and travel time. The adjusted analyses in the other UK study[31] showed a significant increase in early and late NM, with every 15 min increase in travel time (adjOR 1.13, 95% CI 1.07 to 1.20) and (adjOR 1.15, 95% CI 1.05 to 1.26) respectively. Subgroup analysis for nulliparous women showed a statistically significant increased risk of early NM associated with every 15 min increase in travel time from home to the OU (adjOR 1.15, 95% CI 1.06 to 1.25). For term births, late (but not early) NM increased significantly with every 15 min increase travel time from home to the OU (adjOR 1.34, 95% CI 1.13 to 1.59).

In one study from the Netherlands,[35] a travel time of 20 min or more was associated with a significant increase in the combined intrapartum, early and late NM[35–37] (≥20 min vs <20 min, adjOR 1.23, 95% CI 1.07 to 1.41). No NM events were reported in the study from Canada.[44] The study from Japan[47] reported that following a median reduction in travel time from 67 min in 2002 to 39 min in 2006 that there was a decrease in NM rate from 1.67 to 1.28, however, no further analyses were presented.

## Perinatal mortality (PM)

Seven studies examined PM, one from France,[32] one from the Netherlands[38] and five from Canada.[42–46] The French study[32] found no significant association between increasing travel time to the nearest OU and PM based on unadjusted data. However, in the Dutch study a longer travel time (20 min or more) was significantly associated with higher PM[38] (≥20 min vs <20 min, adjOR 1.66, 95% CI 1.59 to 1.74).

The Canadian studies also reported longer travel times to OUs being associated with an elevated risk of PM. A significant increase in PM was reported in women living more than 4 hours away from OUs compared with women living less than 1 hour (>4 hours vs <1 hour adjOR 3.17, 95% CI 1.45 to 6.95).[43] However, findings from the same study suggested no significant increase for women living 1–2 hours and 2–4 hours from an OU compared with those living less than 1 hour from services. Similarly, the PM risk significantly increased in women who lived >1 hour from OUs in a further Canadian study,[44] (cOR 1.54, 95% CI 1.09 to 2.17). When this was divided into different Canadian provinces,[45] the rates of PM were highest in communities living more than 4 hours from an OU in comparison to less than 1 hour in British Colombia only (adjOR 2.84, 95% CI 2.84 to 5.10). Stoll and Kornelsen,[46] found that in women who received midwifery care only, PM was not statistically significantly different for women

**Table 9** Outcomes—travel time

| Outcomes | Study, year, country | Exposure groups | Participants (N, n, %) | | Findings | | | |
|---|---|---|---|---|---|---|---|---|
| **Maternal outcomes** | | | | | | | | |
| Maternal mortality (MM) | No studies | | | | | | | |
| Caesarean section (CS)(overall or intrapartum) | Grzybowski et al[43], 2011, Canada | **Travel time (hrs):<1 1-2 2-4 >4** | | | **All CS** | | | **All CS** |
| | | | **N (35,429)** | **All CS n(%)** | **Groups** | **Crude OR (95%CI)** | **Adjusted OR (95%CI)** | |
| | | | 32814 | 8597 (26.2) | <1 | 1 | 1 | NR |
| | | | 1359 | 313 (23) | 1-2 | 0.84 (0.74, 0.96) | NR | |
| | | | 747 | 156 (20.9) | 2-4 | 0.74 (0.62, 0.89) | NR | |
| | | | 509 | 97 (19.06) | >4 | 0.66 (0.53, 0.83) | NR | |
| | Grzybowski et al[44], 2013, Canada | **Travel time (hrs): <1 >1** | | | **Groups** | **Crude OR (95%CI)** | **Adjusted OR (95%CI)** | **All CS** |
| | | | **N (59 386)** | **n(%)** | | | | **Adjusted OR (95%CI)** |
| | | | 54 714 | 14882 (27.20) | <1 | 1 | 1 | NR |
| | | | 4672 | 1075 (23.01) | >1 | 0.80 (0.75, 0.86) | NR | |
| | Grzybowski et al[45], 2015, Canada | **Travel time (hrs): <1 1-2 2-4 >4** | | | | | | **Alberta CS** |
| | | | **Alberta N (34 453)** | **n%** | **Groups** | **Crude OR (95%CI)** | | **Adjusted OR (95%CI)** |
| | | | 29906 | NR | <1 hour | NR | | 1 |
| | | | 2940 | NR | 1-2 | NR | | 0.86 (0.78, 0.94) |
| | | | 1297 | NR | 2-4 | NR | | 0.67 (0.58, 0.77) |
| | | | 310 | NR | >4 | NR | | 0.64 (0.48, 0.87) |
| | | | **British Columbia N (42,217)** | **n%** | **Groups** | **Crude OR (95%CI)** | | **British Columbia CS** |
| | | | 39 101 | NR | <1 hour | NR | | **Adjusted OR (95%CI)** |
| | | | 1892 | NR | 1-2 | NR | | 1 |
| | | | 623 | NR | 2-4 | NR | | 0.92 (0.83, 1.03) |
| | | | 601 | NR | >4 | NR | | 0.74 (0.61, 0.90) |
| | | | | | | | | 0.70 (0.57, 0.85) |
| | | | **Nova ScotiaN (17 336)** | **n%** | **Groups** | **Crude OR (95%CI)** | | **Nova Scotia CS** |
| | | | 15465 | NR | <1 hour | NR | | **Adjusted OR (95% CI)** |
| | | | 1772 | NR | 1-2 | NR | | 1 |
| | | | 99 | NR | 2-4 | NR | | 0.87 (0.77, 0.98) |
| | | | | | >4 | NR | | 0.67 (0.40, 1.10) |
| | | | | | | | | - |
| | Stoll et al[46], 2014, Canada | **Travel time (hours): <1 1-2 >2** | | | **Groups** | **Crude OR (95% CI)** | | **Adjusted OR (95% CI)** |
| | | | **N (3692)** | **n (%)** | <1 | 1 | | NR |
| | | | 3438 | 633 (18.41) | 1-2 | 1.23 (0.80, 1.91) | | NR |
| | | | 124 | 27 (21.80) | >2 | 1.11 (0.71, 1.72) | | NR |
| | | | 130 | 26 (20.0) | | | | |
| | Darling et al[42], 2019, Canada | **Travel time (mins): ≤ 30 >30** | | | **Groups** | **Crude OR (95%CI)** | | **Adjusted OR (95% CI)** |
| | | | **N** | **n (%)** | ≤ 30 | 1 | | NR |
| | | | 9189 | 536 (5.83) | >30 | 0.74 (0.59, 0.92) | | NR |
| | | | 2236 | 98 (4.44) | | | | |

Continued

**Table 9** Continued

| Outcomes | Study, year, country | Exposure groups | Participants (N, n, %) Groups | N | n (%) | Findings Groups | Crude OR (95% CI) | Adjusted OR (95% CI) |
|---|---|---|---|---|---|---|---|---|
| **Emergency CS** | Grzybowski et al[44], 2014, Canada | **Travel time (hours):** <1 >1 | Groups | N (59 386) | n (%) | Groups | Crude OR (95% CI) | Adjusted OR (95% CI) |
| | | | <1 | 54,714 | 9247 (16.99) | <1 | 1 | NR |
| | | | >1 | 4672 | 701 (15.00) | >1 | 0.80 (0.75, 0.86) | NR |
| **Severe perineal trauma (3rd or 4th degree tear)** | No studies | | | | | | | |
| **Postpartum haemorrhage** | Stolpe et al[39], 2015, Netherlands | **Travel time (mins):** <45 >45 | Groups | N (54) | n (%) | Groups | Median (range) ml | Adjusted OR (95%CI) |
| | | | <45 | 34 | NR | <45 | 2,000 (1,100–7,000) | NR |
| | | | >45 | 20 | NR | >45 | 2,050 (1,000–6,000) (P=0.9) | NR |
| | Grzybowski et al[44], 2013, Canada | **Travel time (hrs):** <1 >1 | Groups | N (59,386) | n (%) | Groups | Crude OR (95%CI) | |
| | | | <1 | 54 714 | 3064 (5.6) | <1 | 1 | |
| | | | >1 | 4672 | 327 (7.0) | >1 | 1.27 (1.13, 1.43) | |
| **Maternal admission to intensive care unit** | Stolpe et al[39], 2015, Netherlands | **Travel time (mins):** <45 >45 | Groups | N (54) | n (%) | Groups | Crude OR (95%CI) | Adjusted OR (95%CI) |
| | | | ≤45 | 34 | 1 (2.94) | ≤45 | 1 | NR |
| | | | >45 | 20 | 1 (5.0) | >45 | 1.74 (0.10, 29.39) | NR |
| **Maternal blood transfusion** | Stolpe et al[39], 2015, Netherlands | **Travel time (mins):** <45 >45 | Groups | N (54) | n (%) | Groups | Median (range) L | |
| | | | ≤45 | 34 | ≤45 | ≤45 | 0 (0–8) | |
| | | | >45 | 20 | >45 | >45 | 2 (0–8) | |
| **Neonatal outcomes** | | | | | | | | |
| **Stillbirth (SB) (overall or intrapartum)** | Paranjothy et al[31], 2014, UK | **Every 15 min increase in travel time** (continuous variable) | Groups | N (412,827) | SB n (%) | Groups | Crude OR (95%CI) | Adjusted OR (95%CI) |
| | | | All women | 412,827 | 135 (0.03) | All women | 1.29 (1.14, 1.47) | 1.13 (0.98, 1.30) |
| | | | Term births in hospital | 387,429 | 85 (0.02) | Term births only | 1.35 (1.16, 1.57) | 1.36 (1.17, 1.59) |
| | | | Nullips births in hospital | 185,419 | 69 (0.04) | Nullips only | 1.33 (1.13, 1.57) | 1.21 (1.02, 1.44) |
| | Combier et al[32], 2013, France | **Travel time (mins):** ≤15 16-30 31-45 ≥46 | Groups | N (111,001) | SB n (%) | Groups | Crude OR (95%CI) | Adjusted OR (95%CI) |
| | | | ≤15 | 70,427 | 333 (0.47) | ≤15 | 1 | 1 |
| | | | 16-30 | 31,792 | 148 (0.47) | 16-30 | 0.98 (0.81, 1.20) | 1.16 (0.96, 1.40) |
| | | | 31-45 | 8445 | 50 (0.59) | 31-45 | 1.25 (0.93, 1.69) | 1.31 (0.89, 1.93) |
| | | | ≥46 | 337 | 3 (0.89) | ≥46 | 1.89 (0.60, 5.92) | 1.90 (0.70, 5.15) |
| | Grzybowski et al[44], 2013, Canada | **Travel time (hrs):** <1 >1 | Groups | N (59,386) | SB n (Rate/1000) | Groups | Crude OR (95%CI) | Adjusted OR (95%%CI) |
| | | | <1 | 54,714 | 274 (5.0) | <1 | 1 | 1 |
| | | | >1 | 4672 | 28 (6.0) | >1 | 1.20 (0.81, 1.77) | NR |

Continued

**Table 9** Continued

| Outcomes | Study, year, country | Exposure groups | Participants (N, n, %) | | Findings | | | |
|---|---|---|---|---|---|---|---|---|
| | | | Groups | N (28,7993) | Early NM (0-6 days)n (%) | Early NM (0-6 days) Groups | Crude OR (95%CI) | Adjusted OR (95%CI) |
| Neonatal mortality (NM) | Dummer et al[29], 2004, UK | Travel time (mins): ≤ 17 17-35 >35 | ≤ 17 | NR | 1850 (NR) | ≤ 17 | NR | 1 |
| | | | 17-35 | NR | 789 (NR) | 17-35 | NR | 0.97 (0.89, 1.06) |
| | | | >35 | NR | 196 (NR) | >35 | NR | 0.95 (0.81,1.1) |
| | | | Groups | N (28,7993) | NM (0-27days)n (%) | NM (0-27 days) Groups | Crude OR (95%CI) | Adjusted OR (95%CI) |
| | | | ≤ 17 | NR | 1854 (NR) | ≤ 17 | NR | 1 |
| | | | 17-35 | NR | 946 (NR) | 17-35 | NR | 0.96 (0.89, 10.4) |
| | | | >35 | NR | 239 (NR) | >35 | NR | 0.95 (0.83, 1.09) |
| | | | Groups | N (28,7993) | Post NM(28-1yr)n (%) | Post NM (28 days – 1yr) Groups | Crude OR (95%CI) | Adjusted OR (95%CI) |
| | | | ≤ 17 | NR | 961 (NR) | ≤ 17 | NR | 1 |
| | | | 17-35 | NR | 400 (NR) | 17-35 | NR | 0.97 (0.86,10.9) |
| | | | >35 | NR | 98 (NR) | >35 | NR | 0.95 (0.77, 1.17) |
| | Paranjothy et al[31], 2014, UK | Every 15 min increase in travel time (continuous variable) | Groups | N | Early NM n (%) | Early NM (0-6 days)Groups | Crude OR (95%CI) | Adjusted OR (95%CI) |
| | | | All women | 412,827 | 609 (0.15) | All women | 1.37 (1.31, 1.45) | 1.13 (1.07, 1.20) |
| | | | Term births only | 387,429 | 177 (0.05) | Term births only | 1.02 (0.86, 1.21) | 0.97 (0.80, 1.17) |
| | | | Nullips only | 185,419 | 303 (0.16) | Nullips only | 1.42 (1.33, 1.51) | 1.15 (1.06, 1.25) |
| | | | Groups | N | Late NM n (%) | Late NM (7-27 days)Groups | Crude OR (95%CI) | Adjusted OR (95%CI) |
| | | | All women | 412,827 | 251 (0.06) | All women | 1.33 (1.23, 1.44) | 1.15 (1.05, 1.26) |
| | | | Term births only | 387,429 | 77 (0.02) | Term births only | 1.24 (1.03, 1.50) | 1.34 (1.13, 1.59) |
| | | | Nullips only | 185,419 | 116 (0.06) | Nullips only | 1.31 (1.15, 1.49) | 1.11 (0.97, 1.28) |

Continued

**Table 9** Continued

| Outcomes | Study,year, country | Exposure groups | Participants (N, n, %) | NM n (%) | Groups | Crude OR (95% CI) | Adjusted OR (95% CI) |
|---|---|---|---|---|---|---|---|
| **NM (Combined intrapartum and early NM)** | Ravelli 2011[35-37], Netherlands | **Travel time (mins):** < 20 mins ≥20 mins | **N (1 054 342)** | **NM n (%)** | **Groups** | **Crude OR (95% CI)** | **Adjusted OR (95% CI)** |
| | | | 558,181 | 789 (0.14) | < 20 mins | 1 | 1 |
| | | | 193,745 | 336 (0.17) | ≥20 mins | 1.23 (1.08, 1.39) | 1.23 (1.07, 1.41) |
| | | | **N (120 896)** | **NM n(/1000)63 (0.05/1000)** | **Low risk women** | **Crude OR (95%CI)** | **Adjusted OR (95%CI)** |
| | | **Low-risk women** | NR | NR | < 20 | NR | 1 |
| | | | NR | NR | ≥20 | NR | 0.8 (0.4, 1.7) |
| | | **Low risk women became high risk during labour** | **N (142,824)** | **NM n(/1000)1.9/1000** | **Low risk women became high risk during labour** | **Crude OR (95%CI)** | **Adjusted OR (95%CI)** |
| | | | NR | NR | <20 | NR | 1 |
| | | | NR | NR | ≥20 | NR | 1.23 (1.04, 1.47) |
| **NM (Combined intrapartum & early & late NM up to 28 days)** | | **Travel time (mins):** <15 15-19 ≥20 | **N (751,926)** | **NM n (1125) (%)** | **Groups** | **Crude OR (95% CI)** | **Adjusted OR (95%CI)** |
| | | | 425,952 | NR | <15 | 1 | 1 |
| | | | 132,229 | NR | 15-19 | 0.97 (0.82, 1.15) | 0.94 (0.79, 1.12) |
| | | | 193,745 | 336 | ≥ 20 | 1.22 (1.07, 1.39) | 1.17 (1.02, 1.36) |
| **NM within 24 hrs** | | | **N (751,926)** | **NM within 24 hrs n (255) (%)** | **Groups** | **Crude OR (95% CI)** | **Adjusted OR (95%CI)** |
| | | | 558,181 | NR | <20 | 1 | 1 |
| | | | 193,745 | NR | ≥ 20 | 1.52 (1.17, 1.97) | 1.51 (1.13, 2.02) |
| **NM 0-7 days** | | | **N (751,926)** | **NM 0-7 dys (523) (%)** | **Groups** | **Crude OR (95% CI)** | **Adjusted OR (95%CI)** |
| | | | 558,181 | NR | <20 | 1 | 1 |
| | | | 193,745 | NR | ≥ 20 | 1.44 (1.20, 1.72) | 1.37 (1.12, 1.67) |
| **NM 8-27 days** | | | **N (751,926)** | **NM 8-27 dys (58) (%)** | **Groups** | **Crude OR (95% CI)** | **Adjusted OR(95%CI)** |
| | | | 558,181 | NR | <20 | 1 | 1 |
| | | | 193,745 | NR | ≥ 20 | 1.30 (0.74, 2.26) | 1.24 (0.67, 2.27) |
| | Grzybowskiet al[44], 2013, Canada | **Travel time (hrs):** <1 >1 | **N (59,386)** | **NM n (%)** | **Groups** | \multicolumn Late NM age <1 month, no events | |
| | | | 54,714 | 0 | <1 | | |
| | | | 4672 | 0 | >1 | | |
| | Aoshima et al[47], 2011, Japan | **Travel time (mins):** Median 39.09 (2006) Median 66.99 (2002) | **N** | **NM n (Rate/ 1000)** | **Groups** | **Crude OR (95%CI)** | **Adjusted OR (95%CI)** |
| | | | NR | NR (1.28) | 2006 | NR | NR |
| | | | NR | NR (1.67) | 2002 | NR | NR |

Continued

**Table 9** Continued

| Outcomes | Study,year, country | Exposure groups | Participants (N, n, %) | | Findings | | | |
|---|---|---|---|---|---|---|---|---|
| | | | Groups | N | Groups | PM n (%) | Crude OR (95%CI) | Adjusted OR (95%CI) |
| **Perinatal mortality (PM)** | Combier et al[32], 2013, France | **Travel time (mins):** ≤15 16 -30 31-45 ≥46 | **Groups** | **N (110,664)** | **Groups** | **PM n (%)** | **Crude OR (95%CI)** | **Adjusted OR (95%CI)** |
| | | | ≤15 | 70,427 | ≤15 | 452 (0.64) | 1 | 1 |
| | | | 16-30 | 31 792 | 16-30 | 195 (0.61) | 0.96 (0.81, 1.13) | 1.08 (0.90, 1.29) |
| | | | 31-45 | 8445 | 31-45 | 59 (0.7.0) | 1.09 (0.83, 1.43) | 1.18 (0.86, 1.62) |
| | | | ≥46 | 337 | ≥46 | 4 (1.19) | 1.86 (0.69, 5.01) | 1.85 (0.66, 5.19) |
| | Ravelli et al[38], 2012, Netherlands | **Travel time (mins):** <20 ≥20 | **Groups** | **N (1,242,725)** | **Groups** | **PM n (Rate/1000)** | **Crude OR (95%CI)** | **Adjusted (OR 95%CI)** |
| | | | <20 | 1,006,607 | <20 | 81 (0.08) | 1 | 1 |
| | | | ≥20 | 236,118 | ≥ 20 | 19 (0.08) | 1.53 (1.47,1.50) | 1.66 (1.59,1.74) |
| | Grzybowski et al[43], 2011, Canada | **Travel time (hrs):** <1 1-2 2-4 >4 | **Groups** | **N (35,429)** | **Groups** | **PM n (Rate/1000)** | **Crude OR (95%CI)** | **Adjusted OR (95%CI)** |
| | | | <1 | 32,814 | <1 | 197 (6.0) | 1 | 1 |
| | | | 1-2 | 1359 | 1-2 | 8 (6.0) | 0.98 (0.48, 1.99) | 1.04 (0.48, 2.22) |
| | | | 2-4 | 747 | 2-4 | 4 (5.0) | 0.89 (0.33, 2.40) | 0.92 (0.33, 2.53) |
| | | | >4 | 509 | >4 | 9 (18.0) | 2.98 (1.52, 5.85) | 3.17 (1.45, 6.95) |
| | | | **PM (SB & early NM)** | | | | | |
| | Grzybowski et al[44], 2013, Canada | **Travel time (hrs):** <1 >1 | **Groups** | **N (59,386)** | **Groups** | **PM n (Rate/1000)** | **Crude OR (95%CI)** | **Adjusted (OR 95%CI)** |
| | | | < 1 | 54 714 | <1 | 383 (7.0) | 1 | NR |
| | | | >1 | 4672 | >1 | 37 (8.0) | 1.54 (1.09, 2.17) | NR |
| | Grzybowski et al[45], 2015, Canada | **Travel time (hrs):** <1 1-2 2-4 >4 | | | **PM (SB & earlyNMAlberta** | | | |
| | | | **Groups** | **Alberta (N=34,453)** | **Groups** | **PM n (%)** | **Crude OR (95%CI)** | **Adjusted OR (95%CI)** |
| | | | <1 hr | 29,906 | <1 | NR | 1 | 1 |
| | | | 1-2 | 2940 | 1-2 | NR | NR | 1.50 (1.03, 2.18) |
| | | | 2-4 | 1297 | 2-4 | NR | NR | 1.35 (0.77, 2.38) |
| | | | >4 | 310 | >4 | NR | NR | 1.40 (0.44, 4.39) |
| | | | | | | | | **PM (SB & early NM) BC** |
| | | | **Groups** | **British Columbia (N=42,317)** | **Groups** | **PM n (%)** | **Crude OR (95%CI)** | **Adjusted OR (95%CI)** |
| | | | <1 hr | 39,101 | <1 | NR | 1 | 1 |
| | | | 1-2 | 1892 | 1-2 | NR | NR | 0.79 (0.43, 1.45) |
| | | | 2-4 | 623 | 2-4 | NR | NR | 1.33 (0.59, 3.01) |
| | | | >4 | 601 | >4 | NR | NR | 2.84 (1.58, 5.10) |
| | | | | | | | | **PM (SB & early NM) Nova Scotia** |
| | | | **Groups** | **Nova Scotia (N= 17,336)** | **Groups** | **PM n (%)** | **Crude OR (95%CI)** | **Adjusted (OR 95%CI)** |
| | | | <1 hr | 15,465 | <1 | NR | 1 | 1 |
| | | | 1-2 | 1772 | 1-2 | NR | NR | 0.66 (0.38, 1.14) |
| | | | 2-4 | 99 | 2-4 | NR | NR | NR |
| | | | >4 | 0 | >4 | NR | NR | NR |

**Table 9** Continued

| Outcomes | Study, year, country | Exposure groups | Groups | N | Outcome n (%) | Groups (Findings) | Crude OR (95% CI) | Adjusted OR (95% CI) |
|---|---|---|---|---|---|---|---|---|
| | Stoll et al[46] 2014, Canada | Travel time (hrs): <1 1-2 >2 | <1 | N (3,692) 3438 | PM n (%) 15 (0.4) | <1 | 1 | NR |
| | | | 1-2 | 124 | 0 | 1-2 | 0.89 (0.05, 14.91) | NR |
| | | | >2 | 130 | 2 (1.5) | >2 | 3.57 (0.81, 15.76) | NR |
| | | PM (SB and early neonatal death up to 7 days) | | | | | | |
| | Darling et al[44] 2019 (42), Canada | Travel time (mins): ≤30 >30 | ≤30 | N (10 681) NR | PM n (%) NR | ≤30 | NR | NR |
| | | | >30 | NR | NR | >30 | NR | NR as RR 2.2 (0.67, 7.43) |
| **Infant mortality (IM)** | | *IM (age 1–12 month)* | | | | | | |
| | Grzybowski et al[44], 2013, Canada | *Travel time (hrs): <1 >1* | <1 | N (59 386) 54 714 | IM n (rate/1000) 109/2.0 | <1 | 1 | NR |
| | | | >1 | 4672 | 14 (3.0) | >1 | 1.51 (0.86, 2.63) | NR |
| **Born before arrival (BBA)** | Combier et al[32], 2013, France | Travel time (mins): <15 15–29 30–44 >45 | <16 | N (111 001) 70 427 | BBA n (%) 132 (0.19) | <16 | 1 | 1 |
| | | | 16–30 | 31,792 | 93 (0.29) | 16–30 | 1.56 (1.20, 2.04) | 1.73 (1.23, 2.46) |
| | | | 31–45 | 8445 | 29 (0.34) | 31–45 | 1.84 (1.23, 2.75) | 1.64 (1.06, 2.54) |
| | | | >45 | 337 | 0 | >45 | - | - |
| | Renesme 2013 (34), France | Travel time (mins): <15 15–29 30–44 >45 | <15 | CasesN (73) (%) 22 (30.2) | ControlN (148) (%) 59 (39.9) | <15 | 1 | 1 |
| | | | 15–29 | 33 (45.2) | 64 (43.2) | 15–29 | 1.79 (0.87, 3.68) | 1.92 (0.86, 4.96) |
| | | | 30–44 | 9 (12.3) | 18 (12.2) | 30–44 | 1.68 (0.58, 4.87) | 1.10 (0.35, 3.48) |
| | | | >45 | 9 (12.3) | 7 (4.7) | >45 | 5.89 (1.12, 30.89) | 6.18 (1.33, 8.65) |
| | Nguyen et al[33], 2016, France | Travel time (mins): ≤ 20 > 20 | >20 | N (188) 94 controls | BBA n (%) 22 (23.4) | >20 controls | 1 | NR |
| | | | >20 | 94 cases | 27 (28.7) | >20 cases | 1.3 (0.7, 2.6) | NR |
| | Engjom et al[41], 2017, Norway | Travel time (hrs): <1 1-2 >2 | 1 | N (646 898) 615 896 | BBA n (%) 3488 (0.60) | 1 | 1 | 1 |
| | | | 1-2 | 25,494 | 844 (3.31) | 1-2 | 6.01 (5.57, 6.49) | NR reported as RR* 5.3 (5.0,5.8) |
| | | | >2 | 5508 | 246 (4.50) | >2 | 8.21 (7.19, 9.37) | NR as RR* 7.2 (6.3,8.2) |
| | Grzybowski et al[43], 2011, Canada | Travel time (hrs): <1 1-2 2-4 >4 | <1 | N (35 429) 32 814 | BBA n (%) 66 (0.20) | <1 | 1 | 1 |
| | | | 1-2 | 1359 | 31 (2.30) | 1-2 | 11.58 (7.53, 17.81) | 6.41(3.69,11.28) |
| | | | 2-4 | 747 | 3 (0.3) | 2-4 | 2.00 (0.63, 6.38) | 0.92 (0.22, 3.88) |
| | | | >4 | 506 | 7 (1.4) | >4 | 6.96 (3.18, 15.25) | 3.63 (1.40, 9.40) |
| | Grzybowski et al[44], 2013, Canada | Travel time (hrs): <1 >1 | <1 | N (59 386) 54 714 | BBA n (%) 164 (0.3) | <1 | 1 | NR |
| | | | >1 | 4672 | 70 (1.5) | >1 | 5.06 (3.82, 6.70) | NR |

Continued

**Table 9** Continued

| Outcomes | Study, year, country | Exposure groups | Participants (N, n, %) | | Findings | | | |
|---|---|---|---|---|---|---|---|---|
| | | | Groups | N (751 926) | Event n (4543) (%) | Groups | Crude OR (95% CI) | Adjusted OR (95% CI) |
| Combined mortality and or Apgar < 4 at 5 mins and or transfer to NICU | Ravelli 2011[35–37], Netherlands | Travel time (mins): <15 15-19 ≥20 | <15 | 425 952 | NR | <15 | 1 | 1 |
| | | | 15-20 | 132 229 | NR | 15-20 | 0.99 (091, 1.07) | 1.11 (1.02, 1.21) |
| | | | ≥20 | 193 745 | NR | ≥20 | 1.11 (1.04, 1.19) | 1.27 (1.17, 1.38) |
| **Neonatal Unit admission (NNU)** | Grzybowski et al[43], 2011, Canada | Travel time (hrs): <1 1–2 2–4 >4 | **NICU level 2 admissions per 1000 births (2001–2004)** | | | | | **NICU 2** |
| | | | Groups | N (35 429) | NICU2 n (rate/1000) | Groups | Crude OR (95% CI) | Adjusted OR (95%CI) |
| | | | <1 | 32 814 | 1082 (33.0) | <1 | 1 | 1 |
| | | | 1-2 | 1359 | 69 (51.0) | 1-2 | 1.57 (1.22, 2.01) | 2.20 (1.59, 3.05) |
| | | | 2-4 | 747 | 8 (11.0) | 2-4 | 0.32 (0.16, 0.64) | 0.31 (0.14, 0.65) |
| | | | >4 | 506 | 14 (27.0) | >4 | 0.83 (0.49, 1.42) | 1.07 (0.54, 2.12) |
| | | | **NICU level 3 per 1000 births (2001–2004)** | | | | | **NICU 3** |
| | | | Groups | N (34 920) | NICU3 n (rate/1000) | Groups | Crude OR (95% CI) | Adjusted OR (95% CI) |
| | | | <1 | 32 814 | 98 (3.0) | <1 | 1 | *NR* |
| | | | 1-2 | 1359 | 11 (8.0) | 1-2 | 2.72 (1.46, 5.09) | *NR* |
| | | | 2-4 | 747 | 4 (5.0) | 2-4 | 1.80 (0.66, 4.90) | *NR* |
| | | | >4 | 509 | 2 (4.0) | >4 | 1.32 (0.32, 5.35) | NR |
| | Grzybowski et al[44], 2013, Canada | Travel time (hrs): <1 >1 | **NICU level 2 (2001–2002 nd 2006–2007) n=74 697** | | | | | **NICU 2** |
| | | | Groups | N (59 386) | NICU2 n (rate/1000) | Groups | Crude OR (95% CI) | Adjusted OR (95% CI) |
| | | | <1 | 54 714 | 1751 (32.0) | <1 | 1 | NR |
| | | | >1 | 4672 | 154 (33.0) | >1 | 1.03 (0.87, 1.22) | NR |
| | | | **NICU level 3 (2001–2002 and 2006–2007) n=74 697** | | | | | **NICU 3** |
| | | | Groups | N | NICU3 n (rate/1000) | Groups | Crude OR (95% CI) | Adjusted OR (95% CI) |
| | | | <1 | 54 714 | 219 (4.0) | <1 | 1 | NR |
| | | | >1 | 4672 | 28 (6.0) | >1 | 1.50 (1.01, 2.23) | NR |
| | | | **NICU (2 and 3)** | | | | | **NICU admission** |
| | | | Groups | N (59 386) | NICU n (rate/1000) | Groups | Crude OR (95% CI) | Adjusted OR (95% CI) |
| | | | <1 | 54 714 | 1970 (36.0) | <1 | 1 | NR |
| | | | >1 | 4672 | 182 (39.0) | >1 | 1.09 (0.93, 1.27) | NR |
| | Darling et al[42], 2019, Canada | Travel time (mins): ≤30 >30 | Groups | N (10 687) | NICU n (%) | Groups | Crude OR (95% CI) | Adjusted OR (95% CI) |
| | | | ≤30 | NR | NR | ≤30 | NR | 1 |
| | | | >30 | NR | NR | >30 | NR | Reported as RR 0.6 (0.44, 0.81) |

Continued

**Table 9** Continued

| Outcomes | Study, year, country | Exposure groups | Travel time (mins): ≤30 >30 | Participants (N, n, %) | | Findings | | | |
|---|---|---|---|---|---|---|---|---|---|
| | | | | | Groups | Groups | Apgar <7 n (%) | Crude RR (95% CI) | Adjusted OR (95% CI) |
| Apgar <7 at 5 mins | Darling et al[42], 2019, Canada | | | N (10 578) | ≤30 | ≤30 | NR | NR | 1 |
| | | | | NR | >30 | >30 | NR | NR | NR as RR 1.02 (0.95, 1.10) |
| | | | | Nullips | Nullips | Groups | Apgar <7 n (%) | Crude OR (95% CI) | Adjusted OR (95% CI) |
| | | | | N (4208) | ≤30 | ≤30 | 51 (1.5) | 1 | NR |
| | | | | 3425 | >30 | >30 | 14 (2.3) | 1.53 (0.84, 2.77) | NR |
| | | | | 621 | | | | | |
| | | | | Mullips | Mullips | Groups | Apgar <7 n (%) | Crude OR (95% CI) | Adjusted OR (95% CI) |
| | | | | N (7661) | ≤30 | ≤30 | 30 (0.5) | 1 | NR |
| | | | | 5764 | >30 | >30 | 11 (0.7) | 1.31 (0.66, 2.62) | NR |
| | | | | 1615 | | | | | |
| HIE | No studies reported | | | | | | | | |

*RR, relative risk.
BW, birth weight; HIE, hypoxic-ischaemic encephalopathy; NICU, neonatal intensive care unit; NR, not reported; Nullips, nulliparous.

living more than 2 hours away from an OU compared with women living less than 1 hour from an OU based on an unadjusted analysis (cOR 3.57, 95% CI 0.81 to 15.76). In Darling et al,[42] the PM rates were not statistically significantly different for women with a planned home birth and more than 30 min drive from hospital (adjRR 2.2, 95% CI 0.67 to 7.43).

### Infant mortality (IM)

One Canadian study[43] reported no significant difference in IM rates for women living less than 1-hour travel time to OU compared with more than 1-hour travel time to OU (cOR 1.51, 95% CI 0.86 to 2.63).

### Born before arrival (BBA)

Six studies reported this outcome, four cohort studies[32 41 43 44] and two case–control studies.[33 34] Five of the six studies found some association between travel time and BBA, four based on adjusted analyses.

There were three studies conducted in France.[32–34] Combier et al,[32] reported that a travel time greater than 15 min was significantly associated with an increased risk of BBA (16–30 min vs <16 min, adjOR 1.73, 95% CI 1.23 to 2.46); (31–45 min vs <16 min, adjOR 1.64, 95% CI 1.06 to 2.54).[32] In a case–control study,[34] the BBA rate increased sixfold when the travel time increased to more than 45 min from home to the OU compared with women who travelled less than 15 min (>45 min vs >15 min, adjOR 6.18 95% CI 1.33 to 28.65). However, in the other case–control study the risk of BBA was not significantly increased in women who travelled for greater than 20 min.[33]

In a study from Norway,[40 41] the risk of BBA increased significantly with longer travel time to the nearest OU from home. Women who travelled more than 2 hours had an eight fold increased risk of BBA compared with women who lived within 1 hour of the nearest OU (>2 hours vs <1 hour, cOR 8.21, 95% CI 7.19 to 9.37).[41]

The studies from Canada[43 44] found a significant increase in BBA in women living in communities greater than 1-hour travel time from an OU compared with those living less than 1 hour away. In Grzybowski et al,[43] women who lived 1–2 hours from an OU had the highest risk of BBA compared with less than 1 hour (adjOR 6.41, 95% CI 3.69 to 11.28) and women who lived greater than 4 hours away also had an increased risk compared with those living less than 1 hour away (adjOR 3.63, 95% CI 1.40 to 9.40); however, there was no difference between those who lived 2–4 hours from an OU and those living less than 1 hour away (adjOR 0.92, 95% CI 0.22 to 3.88). Gryzbowski et al,[44] found a five-=fold increase in BBA in women who lived more than an hour away from an OU in comparison to women who lived less than an hour away (cOR 5.06, 95% CI 3.82 to 6.70).

### Neonatal unit (NNU) admission

Three studies from Canada reported on NNU admission.[42–44] The two studies from British Columbia[43 44]

reported NNU depending on whether the admission was for level 2 care (high dependency) or level 3 care (intensive care). Findings from one of these studies[43] showed NNU level 2 admission increased significantly in babies born to women living more than 1 hour away from an OU compared with less than 1 hour (adjOR 2.20, 95% CI 1.59 to 3.05). For those living 2–4 hours away, level 2 admissions were significantly lower compared with those living less than 1 hour away (adjOR 0.31, 95% CI 0.14 to 0.65). For those living more than 4 hours away, there appeared to be no increase in NNU level 2 admission. For level 3 NNU admission, a significantly increased risk was found for the 1–2 hours category (1–2 hours vs <1 hour, cOR 2.72, 95% CI 1.46 to 5.09). For the other two categories, 2–4 and >4 hours, neither crude nor adjusted analyses showed any significant difference. The number of women in each group decreased with increasing time from an OU. In Grzybowski et al,[44] there was no increased risk of admission to NNU level 2 in babies born to women living more than 1 hour from an OU compared with less than 1 hour, however, admission to NNU level 3 was significantly higher (cOR 1.50, 95% CI 1.01 to 2.23). The third Canadian study from Ontario[42] showed a lower relative risk of NNU admission for planned home births with a travel time greater than 30 min when compared with less than 30 min (adjRR 0.6, 95% CI 0.44 to 0.81).

### Apgar score

Two studies reported on Apgar Score; one from Canada and one from the Netherlands.[37 42] In the Canadian study,[42] no significant difference was found for Apgar score <7 at 5 min between women who planned home birth and lived more or less than 30 min away from an OU, either for nulliparous or multiparous subgroups (adjRR 1.02, 95% CI 0.95 to 1.10).

The study from the Netherlands[37] used a composite outcome of mortality and/or Apgar <4 at 5 min and/or transfer to NNU, and showed a small but significant increase in this outcome in women whose travel time to an OU exceeded 15 min (15–20 min vs <15 min, adjOR 1.11, 95% CI 1.02 to 1.21 and ≥20 min vs <15 min, adjOR 1.27, 95% CI 1.17 to 1.38).

### Hypoxic-ischaemic encephalopathy (HIE)

No studies reported this outcome.

### DISCUSSION

This review describes studies which have explored the associations between OU closure, distance or travel time to an OU, and maternal and neonatal outcomes. The included studies were conducted in the UK, France, the Netherlands, Norway, Canada and Japan. Many studies were from parts of the world where service configuration varied and the study populations were sometimes dispersed over a large geographical area. The included studies differed in their design, geographical boundaries, outcomes measures used and included a wide

range of travel time/distance thresholds used. In addition, although many studies reported that potential confounders were adjusted for in their analyses, many of the outcomes of interest for this review were crude measures of effect without adjustment. Therefore, comparing these studies with each other was a challenge.

All of these studies were brought together to explore whether women who had to travel longer and further to their planned OU were at increased risk of adverse outcomes. There was one reasonably consistent finding which was that there appeared to be an increased risk of BBA the longer it took to reach the OU. This may have been associated with an increased risk for the baby with a suggestion of an increased risk of perinatal or NM in some studies, however, this effect was not consistent across all the studies. There was also an increase in CS rates following closure of an OU and with shorter travel distance and time, however, it is unclear if the difference was related to the exposure or unmeasured differences in CS rates.

### Strengths and limitations of the review

This work is the first to synthesise systematically the current evidence relevant to OU closure and the impact of travel time and travel distance on maternal and neonatal outcomes. Rigorous systematic review methodology was applied, including a sensitive search strategy to identify all the relevant literature, and thorough assessment of potential risks of bias. All screening, data extraction and risk of bias assessment were performed independently by at least two reviewers.

The process of selecting studies for inclusion was challenging due to a lack of reporting of some details, for example, it was not always clear which level of maternity services the study referred to, in others, findings related to the impact of travel time and distance were not always presented despite this being described as a study objective.

### INTERPRETATION OF FINDINGS

It is difficult to conclude from this review whether reconfiguration of maternity services, with closure of OUs, resulting in increased travel distances and times for women is unequivocally associated with worse outcomes for the mother or the baby. Assessing the impact of OU closure and prolonged travel time and distance is not straightforward; to isolate the impact of the closure and travel time and distance on maternal and neonatal outcomes we need to fully understand the models of maternity care, transport services, landscape characteristics, women's satisfaction with care and places of birth available to women in that specific geographical area. Understanding how services are delivered to women is vital when assessing the impact of travel distance and time as services may be adapted to meet the challenges for women living in remote areas, for example by transferring women antenatally a few weeks before birth. Some studies found an increase in CS rates with shorter

distance/travel time. Attributing this solely to closure or reconfiguration of services is problematic as simple analytical comparisons of rates before and after changes do not account for underlying time trends. Future studies might want to consider an interrupted time series design as a more appropriate method.

There remains an urgent need to evaluate the impact of changing maternity service provision. The imperative to close and consolidate OUs into larger units is based on a belief that this will improve safety for both mother and baby. If increasing travel times and distances increases risks to mothers and babies, then the postulated benefits of larger OUs could be offset by the harms of the reconfiguration.

Waiting for closure of OUs to prospectively evaluate the impact on the surrounding maternity population will always be challenging. However, exploring the existing impact of distance and travel time from home to an OU may be a reasonable approach to explore what the impact of reconfiguration may be for a proportion of the women in the area served by the OU which would have these parameters increased by closure of one of more local OU(s). Such a study would need to be large to explore the impact of travel time and distance on substantive harms such as mortality for the baby, so will almost certainly need to use routinely collected data to obtain large numbers. Such studies will also need to include vigorous evaluation of confounders, such as maternal characteristics, socioeconomic status and maternal medical history, which are known to influence birth outcomes; controlling for these factors is vital to determine the OU closure impacts. These studies should also collect data at multiple time points after the closure and apply statistical analysis which considers time-varying relationships and the outcomes.

Measurement of travel time and distance from the woman's place of residence to an OU would also need more sophisticated approaches than previously used in many studies; for example the use of web-based route planners and adjustment for travel conditions rather than using straight line distances or relying on self-reports.

Many study designs assume that travel time and distance have a constant effect on outcomes. If local OUs are far away, it is possible that women will modify their behaviour in relation to when they set off for their OU in labour, if they know they have an hour's journey compared with a 20 min journey. The extent to which this will mitigate the effects of longer travel times would not be seen in a study looking at existing travel times and distances.

## CONCLUSION

Given the substantial variation across studies we were unable to draw firm conclusions regarding the association between OU closure, travel distance or time to obstetric services and maternal and neonatal outcomes. There appears to be a consistent association with BBA with increasing distance and travel time to an OU and a suggestion of increasing risk to the baby. However, few studies have rigorously controlled for potential confounders.

**Acknowledgements** Our thanks to Pamela White for contacting NHS Trusts and obtaining papers, Nia Roberts for development of the search strategy and Mark Willett and Fiona Mackie for providingun published data.

**Collaborators** NA.

**Contributors** PB, FA, RSM and JD conceived the research. All authors developed the protocol and RSM developed the search strategy. RSM, CT, AP, FA and JH screened the search results and full papers. RSM, CT, JH, FA and CO assessed the quality of included papers, extracted the data and synthesised the results. RSM and FA drafted the manuscript and all authors agreed the final manuscript.

**Funding** This research is funded by the National Institute for Health Research (NIHR) Policy Research Programme, conducted through the Policy Research Unit in Maternal Health and Care, 108/0001.

**Disclaimer** The views expressed are those of the author(s) and not necessarily those of the NIHR or the Department of Health and Social Care.

**Competing interests** None declared.

**Patient consent for publication** Not required.

**Provenance and peer review** Not commissioned; externally peer reviewed.

**Data availability statement** All data relevant to the study are included in the article or uploaded as online supplemental information. All the data included in this systematic review are in the public domain.

**ORCID iDs**
Reem Saleem Malouf http://orcid.org/0000-0002-0673-5126
Charles Opondo http://orcid.org/0000-0001-8155-4117

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
