## [Reviewer comments · BMJ Open]

ARTICLE DETAILS

TITLE (PROVISIONAL)	Title: The impact of obstetric unit closures, travel time and distance to obstetric services on maternal and neonatal outcomes in high-income countries: a systematic review
AUTHORS	Malouf, Reem; Tomlinson, Claire; Henderson, Jane; Opondo, Charles; Brocklehurst, Peter; Alderdice, Fiona; Phalguni, Angaja; Dretzke, Janine

VERSION 1 – REVIEW

REVIEWER	Jennifer Hutcheon University of British Columbia
REVIEW RETURNED	17-Feb-2020

GENERAL COMMENTS	This study reported the findings of a systematic review on the consequences of Obstetrical Unit closures, travel time, and travel distance on maternal and newborn health outcomes. It is the first systematic review of this important but challenging topic, and provides a valuable summary of the literature in this field. The authors are to be congratulated on taking a broader, if more labour intensive, inclusion strategy (13,271 articles screened). Although the authors were unable to draw firm conclusions about the association between OU closure, travel distance, or time to obstetric services on maternal and newborn health, I believe it is nevertheless an important contribution to the literature by providing a repository of available knowledge in this field, and highlighting methodological challenges that need to be overcome in future research. Most of my comments relate to the evaluation of obstetrical unit closures: 1) Table 1 contains a mixture of studies- some that directly evaluated closures, others that appear to have compared rates in regions with higher or lower degrees of centralisation. These studies provide different levels of evidence about the effect of closures, and I believe that better highlighting these differences in study types would be helpful. In Table 1A, the authors appear to have directly cut-and-paste the description of the exposure from the original manuscripts into the 'Description of exposure' column, I would suggest instead summarising the exposure definition using their own words to improve consistency and ease of comparison between study types (e.g., 'before and after comparison of closures', 'comparison of regions according to degree of centralisation' etc). I would in fact suggest avoiding cutting-and-pasting for several other columns in the table, but instead summarise the study objectives, etc in a consistent and systematic way across studies. 2) I am concerned about the authors' assessment of the potential for bias in studies evaluating obstetrical unit closures. A central
--

	consideration in studies comparing outcome rates before vs after an intervention (such as an OU closure) is that bias may be introduced if they fail to account for underlying time trends in the outcome rates (e.g., Arch Pediatr Adolesc Med. 2011;165(5):419-423). For example, rates of post-partum haemorrhage are currently rising internationally, independently of OU closure status (e.g., BJOG 2012;119:306, BMC Pregnancy and Childbirth 2009;55). For such outcomes, a simple comparison of rates before vs after a closure will show higher rates post-closure that could incorrectly be attributed to the closure. To avoid this bias, more robust study designs such as interrupted times series analyses or difference in difference analyses are needed to isolate the effect of the closure from underlying trends (IJE 2017;348; Family Practice 2000; 17: S11–S18). Yet, the two studies that employed such methods (the difference-in-difference designs of Grytten et al and Hutcheon et al) were rated as having a high risk for bias, and concluding that ‘only one of ten studies reporting that temporal variation was adjusted for in the analysis’. I would suggest revisiting the criteria for evaluating this source of bias. 3) On a related note, quasi-experimental designs such as time-series analyses and difference-in-difference analyses inherently control for potential confounders such as maternal characteristics and comorbidities through their design. By using each site as its own control, or isolating the effect of the change immediately at the time of the closure, the designs provide a quasi-randomization of patients, preventing confounding so long as there were no abrupt changes in maternal characteristics of women in the catchment area immediately at the same time as the closure (which is unlikely). This should be reflected in the risk of bias assessment for ‘Potential confounders adjusted for and listed’ column. 4) The authors correctly evaluated whether the closures were independent of other changes over time (second column from the right in Table 1B). However, this issue appears to have been conflated with the issue of underlying time trends in the outcome noted above (e.g., comment in Table 1B for Grytten paper that risk was low because “temporal variation adjusted for in analysis”). These are two distinct sources of bias: one refers to change over time in the outcome (for which controlling for calendar time is needed), while temporal changes in other things requires control for the relevant covariates. These should be evaluated as two separate sources of bias. 5) I agree with the authors’ conclusions that there remains an urgent need to evaluate the impact of changing maternity service provision. I like the authors’ current discussion of what studies generating this evidence might look like, and would encourage the authors to explore this more: what studies identified in the review can serve as examples of ‘well-done’ studies, what are the key confounders that need to be identified in such studies, etc. If the authors are proposing to conduct studies comparing outcomes of women living with different times/distances to the closest obstetrical services, what is needed to prevent confounding by inherent differences in women who opt to live in more rural vs more urban areas (risk profile related to activity, lifestyle, health care services utilisation, etc that are very challenging to measure accurately)? 6) In my jurisdiction (Canada), many women living in remote areas are transferred antenatally to larger centres several weeks before their due date for delivery. As such, their ‘travel time’ for labour and delivery services does not reflect the distance between their
--	---

	residence and the closest obstetrical services. The authors may wish to discuss the need for this type of context in evaluating studies on this topic. Minor issues 1) Table 1A, column 1, entry for Hemminki study- typo in word 'Finland' and multiple different reference numbers included. 2) While appreciating that the authors pre-specified select maternal and newborn health outcomes in their protocol (and this is therefore the literature that was systematically retrieved), it would be useful to list the other outcomes in the retrieved studies for readers interested in other outcomes (with the caveat of course that it may not be an exhaustive list as these outcomes were not systematically searched for).
--	---

REVIEWER	Dr. María Belén Conesa Ferrer University of Murcia (Spain)
REVIEW RETURNED	03-Mar-2020

GENERAL COMMENTS	First of all, I consider that the main objective of this systematic review is a very interesting topic. I think that one of the inclusion criteria should be the good quality of the study methodology. Most studies in table 1A have no eligibility criteria, participant characteristics, study objectives, and exclusion criteria. In fact, authors comment on page 12, lines 5-6 that only 4 studies reported and used appropriate data analysis. The results from studies with poor design quality have no scientific interest.
---

REVIEWER	Sebastiano Barbieri Centre for Big Data Research in Health, UNSW Australia
REVIEW RETURNED	12-May-2020

GENERAL COMMENTS	This systematic review synthesises current evidence on the effect of obstetric unit (OU) closures on maternal and neonatal outcomes in the surrounding or comparable population and on the association between travel distance or time and maternal and neonatal outcomes, including perinatal mortality and babies born before arrival (BBA). The authors applied a wide and sensitive search strategy to identify relevant studies in high-income countries with universal health coverage comparable to the UK. The selected studies were evaluated in terms of employed methods and risk of bias. The review indicates that there is some evidence of OU closures being associated with BBA increase; however, there was limited or no evidence on the impact of the exposures on the majority of maternal and neonatal outcomes. This is a policy-relevant and generally well-written synthesis of current evidence on the effects of OU closures and travel times to OUs on maternal and neonatal outcomes. The review is methodologically sound. I only have a few minor suggestions for improvement:  • The title should indicate that the review is limited to high-income/OECD countries.
--

	 • The introduction could be expanded with additional background information on some of the analysed outcomes, e.g. consequences of BBA, definition of Apgar score etc. • The aims at the end of page 4 – beginning of page 5 could include some examples of assessed outcomes. • 9 26-30 Review methods: please indicate how the review tasks were allocated to members of the review team. • 9 29 Review methods: please include a reference to the Eppi-reviewer software. • 10 29 Patient and Public Involvement: the first sentence is unclear, please rephrase. • 19 11 Description of included studies: Norway is in Europe, please rephrase. • 27 28 Interpretation of findings: Please explain more in detail (or rephrase) what you mean by “To obtain large numbers these studies will almost certainly include vigorous evaluation of confounders.” It would also be interesting to provide some examples of “more sophisticated approaches” that are required to measure travel times and distances. • URLs in references are not formatted correctly (e.g. dots are missing) • PRISMA Flow Diagram: are the second and third boxes really both n=13,271? Please check. • The paper should be carefully checked for typos (e.g. 4 9 “to synthesis”, 16 54 “0.90l”, 20 10 “an 1 hour”, 22 39 “travel time 20 mins”, 23 33 “outcome.,”). Also check the consistency of the format used to report 95%CI.
--	--

VERSION 1 – AUTHOR RESPONSE

Reviewer: 1	
This study reported the findings of a systematic review on the consequences of Obstetrical Unit closures, travel time, and travel distance on maternal and newborn health outcomes. It is the first systematic review of this important but challenging topic, and provides a valuable summary of the literature in this field. The authors are to be congratulated on taking a broader, if more labour intensive, inclusion strategy (13,271 articles screened). Although the authors were unable to draw firm conclusions about the association between OU closure, travel distance, or time to obstetric services on maternal and newborn health, I believe it is nevertheless an important contribution to the literature by providing a repository of available knowledge in this field, and highlighting methodological challenges that need to be overcome in future research.	We thank the reviewer for their comments.
1) Table 1 contains a mixture of studies- some that directly evaluated closures, others that	We agree with the comments, and have modified the descriptions of the exposure for

appear to have compared rates in regions with higher or lower degrees of centralisation. These studies provide different levels of evidence about the effect of closures, and I believe that better highlighting these differences in study types would be helpful. In Table 1A, the authors appear to have directly cut-and-paste the description of the exposure from the original manuscripts into the 'Description of exposure' column, I would suggest instead summarising the exposure definition using their own words to improve consistency and ease of comparison between study types (e.g., 'before and after comparison of closures', 'comparison of regions according to degree of centralisation' etc). I would in fact suggest avoiding cutting-and-pasting for several other columns in the table, but instead summarise the study objectives, etc in a consistent and systematic way across studies	consistency. For each study we have listed the number and type of obstetric units pre-change and post-change to make it clear whether we are looking only at closures or also at reconfiguration of units (and what type). We have also distinguished between the timing of any service changes and the time periods being compared in the analysis, and added detail on the geographical area affected by the changes.
2) I am concerned about the authors' assessment of the potential for bias in studies evaluating obstetrical unit closures. A central consideration in studies comparing outcome rates before vs after an intervention (such as an OU closure) is that bias may be introduced if they fail to account for underlying time trends in the outcome rates (e.g., Arch Pediatr Adolesc Med. 2011;165(5):419-423). For example, rates of post-partum haemorrhage are currently rising internationally, independently of OU closure status (e.g., BJOG 2012;119:306, BMC Pregnancy and Childbirth 2009;55). For such outcomes, a simple comparison of rates before vs after a closure will show higher rates post-closure that could incorrectly be attributed to the closure. To avoid this bias, more robust study designs such as interrupted times series analyses or difference in difference analyses are needed to isolate the effect of the closure from underlying trends (IJE 2017;348; Family Practice 2000; 17: S11–S18). Yet, the two studies that employed such methods (the difference-in-difference designs of Grytten et al and Hutcheon et al) were rated as having a high risk for bias, and concluding that 'only one of ten studies reporting that temporal variation was adjusted for in the analysis'. I would suggest	Thank you for pointing out the limitation of included studies specifically the studies with a before and after design. We added the following to the interpretation of findings: Some studies found an increase in CS rates with shorter distance/travel time. Attributing this solely to closure or reconfiguration of services is problematic as simple analytical comparisons of rates before and after changes do not account for underlying time trends. Future studies might want to consider an interrupted time series design as a more appropriate method.

revisiting the criteria for evaluating this source of bias.	
3) On a related note, quasi-experimental designs such as time-series analyses and difference-in-difference analyses inherently control for potential confounders such as maternal characteristics and comorbidities through their design. By using each site as its own control, or isolating the effect of the change immediately at the time of the closure, the designs provide a quasi-randomization of patients, preventing confounding so long as there were no abrupt changes in maternal characteristics of women in the catchment area immediately at the same time as the closure (which is unlikely). This should be reflected in the risk of bias assessment for 'Potential confounders adjusted for and listed' column.	Thank you for your comments. We added the following to the interpretation of findings: Such studies will also need to include vigorous evaluation of confounders, such as maternal characteristics, socioeconomic status and maternal medical history, which are known to influence birth outcomes; controlling for these factors is vital to determine the OU closure impacts. These studies should also collect data at multiple time points after the closure and apply statistical analysis which considers time-varying relationships and the outcomes.
4)The authors correctly evaluated whether the closures were independent of other changes over time (second column from the right in Table 1B). However, this issue appears to have been conflated with the issue of underlying time trends in the outcome noted above (e.g., comment in Table 1B for Grytten paper that risk was low because “temporal variation adjusted for in analysis”). These are two distinct sources of bias: one refers to change over time in the outcome (for which controlling for calendar time is needed), while temporal changes in other things requires control for the relevant covariates. These should be evaluated as two separate sources of bias.	The risk of bias criteria related to analysis and confounding have been reviewed and the risk of bias Table 1b revised as follows: Grytten’s study: For “analysis method and appropriate reporting”: it is UNCLEAR. The method of analysis was appropriate, but it is unclear what the coefficients represent and confidence intervals are not reported in the paper. Hutcheon’s study: For analysis method and appropriate reporting: It has been changed from “HIGH” to “LOW”, as the within-community fixed-effects design was used. For “independent of other changes over time” domain it has been changed from “UNCLEAR” to “LOW ”, because of using difference in difference analysis which separates the effect of the closure from underlying time trends of reported outcomes.
5)I agree with the authors’ conclusions that there remains an urgent need to evaluate the impact of changing maternity service provision. I like the authors’ current discussion of what studies generating this evidence might look like,	Thank you for the agreement and positive comments on the interpretation of findings:

and would encourage the authors to explore this more: what studies identified in the review can serve as examples of ‘well-done’ studies, what are the key confounders that need to be identified in such studies, etc. If the authors are proposing to conduct studies comparing outcomes of women living with different times/distances to the closest obstetrical services, what is needed to prevent confounding by inherent differences in women who opt to live in more rural vs more urban areas (risk profile related to activity, lifestyle, health care services utilisation, etc that are very challenging to measure accurately)?	We have added more detail in the discussion relating to what key confounders are likely to be (those known to affect birth outcomes, i.e. maternal characteristics, socioeconomic status and maternal medical history). We have also added a comment on better ways of measuring exposure (e.g. though use of online planners) and study design (i.e. interrupted time series preferable to before and after study).
6) In my jurisdiction (Canada), many women living in remote areas are transferred antenatally to larger centres several weeks before their due date for delivery. As such, their ‘travel time’ for labour and delivery services does not reflect the distance between their residence and the closest obstetrical services. The authors may wish to discuss the need for this type of context in evaluating studies on this topic.	We added the following to the interpretation of findings: Understanding how services delivered to women is vital when assessing the impact of travel distance and time as services may be adapted to meet the challenges for remote areas dwellers, for example by transferring women antenatally a few weeks before birth
7)Table 1A, column 1, entry for Hemminski study- typo in word ‘Finland’ and multiple different reference numbers included.	Corrected.
8) While appreciating that the authors pre-specified select maternal and newborn health outcomes in their protocol (and this is therefore the literature that was systematically retrieved), it would be useful to list the other outcomes in the retrieved studies for readers interested in other outcomes (with the caveat of course that it may not be an exhaustive list as these outcomes were not systematically searched for).	We were extremely inclusive in this review in terms of important maternal and neonatal outcomes. These were decided on in consultation with one of the authors who has considerable experience in obstetrics and maternal/neonatal health (PB) and were pre-specified in our protocol. Studies were included where they reported on at least one of the pre-specified outcomes. Most of the included studies did not report on additional outcomes and there were some pre-specified outcomes that we could not identify any evidence for.
Reviewer: 2	
First of all, I consider that the main objective of this systematic review is a very interesting topic.	Thank you for your comments.
1)I think that one of the inclusion criteria should be the good quality of the study methodology.	It is good systematic review practice to include all evidence. A problem with having an cut-off for study inclusion based on quality is poor reporting – studies may be included or excluded where is it unclear if their methodology was

	appropriate or not. A further role of systematic reviews is to highlight methodological flaws so that a) the evidence can be interpreted in light of any such flaws, and b) methodological recommendations can be made for future studies. We have addressed both of these points extensively. Excluding evidence also leaves a systematic review open to criticism as evidence is then being ignored – and a cut-off for including on the basis of methodological quality or reporting will always be arbitrary to an extent
2) Most studies in table 1A have no eligibility criteria, participant characteristics, study objectives, and exclusion criteria. In fact, authors comment on page 12, lines 5-6 that only 4 studies reported and used appropriate data analysis. The results from studies with poor design quality have no scientific interest.	We revised table 1A and added more information. Most studies do report eligibility criteria, this information is lacking only in those studies reported as abstracts or where we had access to unpublished data. We agree that participant characteristics are often poorly reported and that data analysis methods were also either poorly reported or not appropriate. As mentioned in the above comment, it is one role of systematic reviews to highlight where there is poor reporting and methodology. This is helpful for showing that the evidence to date is poor and that it can therefore not be used to draw firm conclusions or make specific recommendations.
Reviewer: 3	
This systematic review synthesises current evidence on the effect of obstetric unit (OU) closures on maternal and neonatal outcomes in the surrounding or comparable population and on the association between travel distance or time and maternal and neonatal outcomes, including perinatal mortality and babies born before arrival (BBA). The authors applied a wide and sensitive search strategy to identify relevant studies in high-income countries with universal health coverage comparable to the UK. The selected studies were evaluated in terms of employed methods and risk of bias. The review indicates that there is some evidence of OU closures being associated with BBA increase; however, there was limited or no evidence on the impact of the exposures on the majority of maternal and neonatal outcomes.	Thank you for your feedback on the paper.

This is a policy-relevant and generally well-written synthesis of current evidence on the effects of OU closures and travel times to OUs on maternal and neonatal outcomes. The review is methodologically sound. I only have a few minor suggestions for improvement:	
1)The title should indicate that the review is limited to high-income/OECD countries	We have changed the title accordingly.
2)The introduction could be expanded with additional background information on some of the analysed outcomes, e.g. consequences of BBA, definition of Apgar score etc	We have added the following to the introduction: There is a rise in the risk of babies born before arrival (BBA, also referred to as unplanned out of hospital births). Being born before arrival is more common before term and has been reported to be associated with higher perinatal mortality (5). Conversely, Lasswell.etal (6) found neonatal mortality was reduced when services were configured to ensure very preterm infants are born in a large maternity hospital with neonatal intensive care unit (Level III). In addition to mortality, Apgar scores (a standardised measure of the physical condition of a newborn infant) and neonatal admission to intensive care provide an indication of perinatal infant health.
3) The aims at the end of page 4 – beginning of page 5 could include some examples of assessed outcomes.	We added a new paragraph in the introduction and some examples of the outcomes are listed.
4) 9 26-30 Review methods: please indicate how the review tasks were allocated to members of the review team.	We added the initials of the authors.
5) 9 29 Review methods: please include a reference to the Eppi-reviewer software.	We added the citation.
6)10 29 Patient and Public Involvement: the first sentence is unclear, please rephrase.	This sentence has been rephrased to: We involved our Parent, Patient, and Public Involvement (PPPI) Stakeholders Network, to explore which outcomes were important from a maternal perspective.
7) Pag 18. Description of included studies: Norway is in Europe, please rephrase.	This sentence has been rephrased to: ..... were examined in studies from Norway, Japan and Canada compared to studies in other countries (all European).

8)27 28 Interpretation of findings: Please explain more in detail (or rephrase) what you mean by “To obtain large numbers these studies will almost certainly include vigorous evaluation of confounders.” It would also be interesting to provide some examples of “more sophisticated approaches” that are required to measure travel times and distances.	We added the following to the interpretation of findings: Such studies will also need to include vigorous evaluation of confounders, such as maternal characteristics, socioeconomic status and maternal medical history, which are known to influence birth outcomes; controlling for these factors is vital to determine the OU closure impacts. We have also added the following: ...also need more sophisticated approaches than has previously been used in many studies; for example the use of web-based route planners and adjustment for travel conditions rather than using straight line distances or relying on self-reports.
9)URLs in references are not formatted correctly (e.g. dots are missing)	Done.
10) PRISMA Flow Diagram: are the second and third boxes really both n=13,271? Please check	Checked.
11) The paper should be carefully checked for typos (e.g. 4 9 “to synthesis”, 16 54 “0.90l”, 20 10 “an 1 hour”, 22 39 “travel time 20 mins”, 23 33 “outcome.,”). Also check the consistency of the format used to report 95%CI.	Checked.

VERSION 2 – REVIEW

REVIEWER	Dr. María Belén Conesa Ferrer University of Murcia (Spain)
REVIEW RETURNED	06-Aug-2020

GENERAL COMMENTS	First of all, I would like to say that I found reviewing this study very pleasant since you are dealing with a topic of great professional interest. I have minor comments for authors: On ‘Search results section’ (page 12), where authors comment “Figure 1”, I consider that the title where Figure 1 appears should be added and authors should change in the explanation of this Figure the “Reasons for inclusion”, rather than “Reasons for exclusion”. On ‘Interpretation of findings section’, authors explain that studies will also need to include vigorous evaluation confounders because several of them are known to influence birth outcomes. I suggest
--

	that one of them is woman's satisfaction due to the great impact this could have on the health of both the woman and the newborn.
REVIEWER	Sebastiano Barbieri UNSW, Australia
REVIEW RETURNED	04-Aug-2020
GENERAL COMMENTS	All my comments were addressed in a satisfactory manner.

VERSION 2 – AUTHOR RESPONSE

Reviewers' comment (2):

On 'Search results section' (page 12), where authors comment "Figure 1", I consider that the title where Figure 1 appears should be added and authors should change in the explanation of this Figure the "Reasons for inclusion", rather than "Reasons for exclusion".

Authors' responses: We added the following title to Figure 1: "PRISMA flow diagram".

On 'Interpretation of findings section', authors explain that studies will also need to include vigorous evaluation confounders because several of them are known to influence birth outcomes. I suggest that one of them is woman's satisfaction due to the great impact this could have on the health of both the woman and the newborn.

Authors' responses: We added to the interpretation of findings: Woman's satisfaction with care.

Reviewers' comment (3): No further revisions required